# Plant-based red colouration of shell beads 15,000 years ago in Kebara Cave, Mount Carmel (Israel)

**Laurent Davin**[1,2,3]*, **Ludovic Bellot-Gurlet**[4]*, **Julien Navas**[5]

**1** The Hebrew University of Jerusalem, Institute of Archaeology, Jerusalem, Israel, **2** CNRS, UAR 3132 Centre de Recherche Français à Jérusalem (CRFJ), Jerusalem, Israel, **3** CNRS, UMR 8068 Technologie et Ethnologie des Mondes PréhistoriqueS (TEMPS), Nanterre, France, **4** CNRS, UMR 8233 De la Molécule aux Nano-objets: Réactivité, Interactions et Spectroscopies (MONARIS), Sorbonne Université, Paris, France, **5** Conservatoire national des arts et métiers, Paris, France

* laurent.davin.etu@gmail.com (LD); ludovic.bellot-gurlet@sorbonne-universite.fr (LBG)

**Data Availability Statement:** All relevant data are within the paper and its Supporting Information files.

**Funding:** Fyssen Foundation post-doctoral fellowship (LD) https://www.fondationfyssen.fr/en/

## Abstract

Decorating the living space, objects, body and clothes with colour is a widespread human practice. While the habitual use of red mineral pigments (such as iron-oxide, e.g., ochre) by anatomically modern humans started in Africa about 140,000 years ago, the earliest documentation of the use of organic plant or animal-based red pigments is known from only 6,000 years ago. Here, we report the oldest reliable evidence of organic red pigment use 15,000 years ago by the first sedentary hunter-gatherers in the Levant. SEM-EDS and Raman Spectroscopy analyses of 10 red-stained shell beads enabled us to detect and describe the use of a colourant made of Rubiaceae plants roots (*Rubia* spp., *Asperula* spp., *Gallium* spp.) to colour personal adornments from the Early Natufian of Kebara cave, Mount Carmel, Israel. This adds a previously unknown behavioural aspect of Natufian societies, namely a well-established tradition of non-dietary plant processing at the beginning of the sedentary lifestyle. Through a combined multidisciplinary approach, our study broadens the perspectives on the ornamental practices and the *chaînes opératoires* of pigmenting materials during a crucial period in human history.

## Introduction

Humans' perception of the red colour greatly influences their affective, cognitive, and behavioural responses in achievement, affiliation and attraction contexts [1]. As demonstrated by Hill & Barton [2], there is a "red effect" found to be most prominent in males, which states that wearing red enhances one's dominance, aggressiveness, and testosterone level, facilitating competitive positive outcomes. This influence of red colour on the mind of anatomically modern humans probably explains, at least in part, why they started, about 140,000 years ago in Africa [3], to use habitually red mineral pigments, such as iron-oxide (commonly called ochre) in order to decorate their living space, objects, body and clothes. The use of red organic pigments of plant or animal origin, which are brighter, 'purer', and stronger in tinting power than

Irene Levi Sala CARE Archaeological Foundation (LD) https://www.prehistory.org.il/irene-levi-sala-care-archaeological-foundation-grants-in-aid/ Hebrew University of Jerusalem (LD) https://archaeology.huji.ac.il/ Centre de Recherche Français à Jérusalem (LD) https://www.crfj.org/en/ The funders had no role in study design, data collection and analysis, decision to publish, or preparation of the manuscript."

**Competing interests:** The authors have declared that no competing interests exist.

inorganic pigments [4–6] (and therefore more attractive to human eyes [7]), appeared only much later, ca. 6,000 years ago [8–10]. The early use of red organic pigments demonstrates a significant development of the *chaînes opératoires* of pigmenting materials since they were used extensively by many cultures and for many purposes for millennia (e.g. dyeing textiles, painting, make-up, etc.) until the recent invention of synthetic pigments in the mid-19th century [11].

Current prehistoric research recognises red ochre as a universally applied material that serves various purposes, from symbolic and ritual display to utilitarian or functional uses, depending on the contexts [3, 12]. Many of these were recognised in the Natufian archaeological culture (15,000–11,650 cal BP), which marks, in the Levant, the transition from hunter-gatherer Palaeolithic societies into fully-fledged agricultural economies of the Neolithic [13]. The Natufians were the first hunter-gatherers to adopt a sedentary lifestyle, a dramatic economic and societal change associated with growing social complexity [14] as reflected also in various aspects of their material culture involving red ochre: burials where bodies were wrapped in textiles coloured with ochre [15, 16] or bones that were decorated once the body was decomposed as in Azraq 18 [17]; artistic manifestations in the form of anthropomorphic and zoomorphic figurines or decorated objects with incised geometric patterns decorated with ochre [18–20]; personal adornments with thousands of shell, bone and tooth beads ochre coloured [21–23]; bone aerophones coloured with ochre [24] and durable stone-built structures whose lime coating is ochre red [20].

In this paper, we present a plant-based red colourant used in the Early Natufian site of Kebara Cave, north-western Israel, which provides yet new perspectives on the development of ornamental practices and the *chaînes opératoires* of colouring materials during a crucial period in human history.

## Kebara Cave

Kebara cave is located on the western slope of Mount Carmel [25], ca. 60 m above sea level and about 2.5 km from the current shoreline of the Mediterranean Sea (about 8–13 km 15,000 years ago when the sea level was about 80 m lower than today [26]) (Fig 1A and 1B). The 1931 short excavation by F. Turville-Petre and C.A. Baynes identified a long stratigraphic sequence from the Middle Palaeolithic (Mousterian: Layer F) to modern time (Layer A) [27]. Layer B contained an Early Natufian occupation extending over the whole surface of the cave (ca. 300 m²) and ranging in thickness from 0.6 m at the cave's entrance to 2.2 m at its back. A sterile layer separates The Natufian layer from the underlying Kebaran occupation (Layer C) (Fig 1C and 1D). A burnt bone discovered at the bottom of layer B provided one of the oldest dating of the Natufian culture with a date of 12,470 ± 180 BP [25] or 13,315–12,114 cal BC (calibrated in OxCal v.4.4 [28]). Except for the burials of 48 individuals [29], no architectural remains were recognised during the excavation, in contrast with the exceptional richness of the archaeological material remains (partially published in a single short preliminary report [30–32]). Domestic occupation is not in doubt, and the relative abundance of sickle blades in the lithic industry [33], remains of large game in the faunal spectrum [34], and a rich and decorated bone industry (including zoomorphic figurations [35, 36] (Fig 13)) suggest a wide range of activities on-site to which we are adding now also the obtaining/production/processing and use of colouring organic pigments.

## Personal adornments from Kebara Cave

Although presented only briefly in former publications, the personal adornments are indeed exceptional both in variety and craftsmanship and well preserved with more than 400 oval

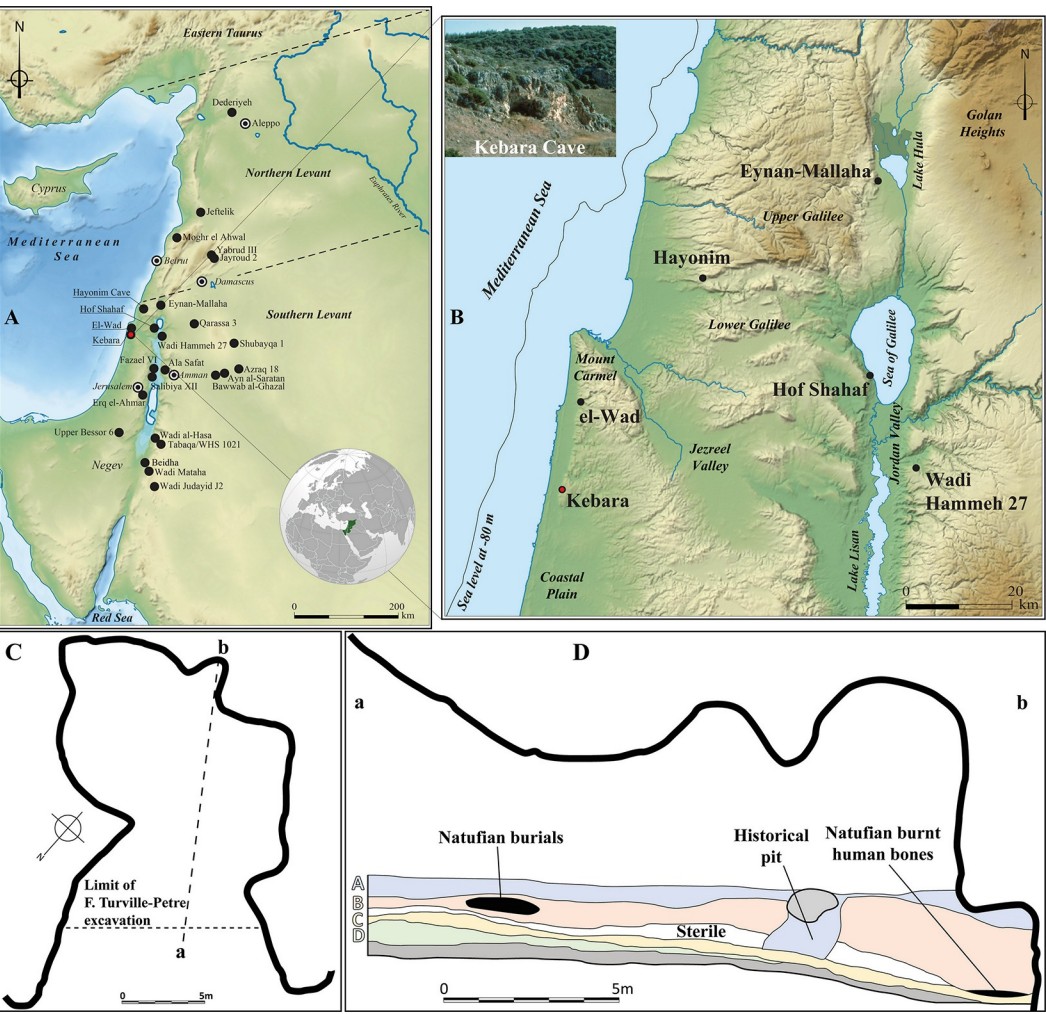

**Fig 1.** A: Map of the distribution of Early Natufian sites in the Levant; B: Detailed map of the region of Kebara cave in the Southern Levant with the position of the shoreline (Mediterranean Sea level at -80m [26]) and the Lake Lisan at 15,000 BP, Background map republished from Wikipedia under a CC BY license, with permission from Ivan d'Hostingue and NordNordWest, original copyright 2023; C: Plan of Kebara Cave (modified after [29]); D: Cross section of the deposits in Kebara Cave identified by F. Turville-Petre (modified after [25]) (CAD L.D.).

bone pendants (Fig 13), 150 of which were discovered in a "cache" near the burials [37], F. Turville-Petre [31] also mentions perforated teeth and shells. However, the dispersal of these hundreds of bone, tooth and shell beads and pendants to various museums in the United Kingdom and North America immediately after the excavation has considerably limited the understanding of these adornments, of which only a small sample has been studied and published to date [38, 39]. It demonstrated, for the first time in a Natufian context, a sophisticated colouration of the oval bone pendants by controlled heating [39] producing a rich variation of shades from white, grey, brown to black (Fig 13). Our study of the only remaining collections in the Levant, stored at the Rockefeller Museum in Jerusalem (which represent only a small proportion of the original adornments assemblage), has allowed us to assess the considerable importance of Kebara cave in understanding the ornamental practices of the Early Natufians. This is notably the case for shell adornments, particularly for Kebara scaphopod beads. While, in other Natufian sites in the Mediterranean zone, the majority of the recovered shells

originate from the nearby Mediterranean Sea [21, 22], in Kebara, we note the existence of acquisition patterns over very long distances and in an intensity that has no equal in other Early Natufian sites in the same geographic zone [40]. This is illustrated by the presence of *Dentalium bisexangulum* beads (Fig 2,3,5,7 and 8) derived from the Red Sea about 400 km to the south [41] and the beads of *Dentalium sexangulum* (Fig 2.10) deriving from Miocene-Pliocene fossil deposits about 400 km further to the north, in the northern Levant [42].

## Colourant matters

In addition to the acquisition patterns over very long distances and the colouration of the bone pendants by controlled heating [39], another primary interest of the personal adornments in Kebara lies in the presence of colouring residues. The colour traces on 16 beads and pendants (Table 1) were studied by non-invasive analytical methods (according to Israel Antiquities Authority guidelines): thirteen shell, two teeth and one bone beads and pendants (Fig 2). The colour residues range from light orange to deep red and have been found preserved in micro-concavities or notches of the internal and/or external surfaces of the beads. Unfortunately, the preserved distribution of the colour staining is insufficient to reconstruct the original

**Table 1. List of the Early Natufian beads and pendants from Kebara Cave analysed in this study.** Origin of the shell beads: MS (Mediterranean Sea), RS (Red Sea), FD (Fossil deposits).

| N° on Fig 2 | Inventory N° | Type | Subtype | Species | Origin | Burnt (Yes/No) | Residue position | Color Residue | Raman identification |
|---|---|---|---|---|---|---|---|---|---|
| 1 | KEB 33.159 | Shell bead | Gastropod bead | *Columbella rustica* | MS | N | Lip | Organic | Rubiaceae |
| 2 | KEB 33.161 | Shell bead | Gastropod bead | *Columbella rustica* | MS | Y | Lip / Tip | | |
| 3 | KEB 33.163.2.3 | Shell bead | Scaphopod bead | *Dentalium bisexangulum* | RS | N | Dmax Internal | | |
| 4 | KEB 337336.4 | Shell bead | Scaphopod bead | *Antalis vulgaris* | MS | N | Dmax Ext/Dmin Int | | |
| 5 | KEB 33.163.2.4 | Shell bead | Scaphopod bead | *Dentalium bisexangulum* | RS | N | Dmin Internal | | |
| 6 | KEB 33.163.1 | Shell bead | Scaphopod bead | *Antalis vulgaris* | MS | N | Dmax Int/Dmin Ext | | |
| 7 | KEB 337336.2 | Shell bead | Scaphopod bead | *Dentalium bisexangulum* | RS | Y | Dmax Int/Ext | | |
| 8 | KEB 337336.1 | Shell bead | Scaphopod bead | *Dentalium bisexangulum* | RS | N | Dmin Ext | | |
| 9 | KEB 33.163.2.1 | Shell bead | Scaphopod bead | *Antalis vulgaris* | MS | N | External | | |
| 10 | KEB 337336.3 | Shell bead | Scaphopod bead | *Dentalium sexangulum* | FD | N | Dmax Ext/Dmin Int | | |
| 11 | KEB I.10720.1 | Shell bead | Scaphopod bead | *Antalis vulgaris* | MS | N | Dmax External | Mineral | Hematite |
| 12 | KEB I.10720.2 | Shell bead | Scaphopod bead | *Antalis vulgaris* | MS | N | Dmax External | | |
| 13 | KEB I.10720.3 | Shell bead | Scaphopod bead | *Antalis vulgaris* | MS | N | Dmax External | | |
| 14 | KEB 33.83 | Tooth Pendant | Incisive Pendant | *Large ungulate* | - | N | Root | | |
| 15 | KEB 33.84 | Tooth Pendant | Canine Pendant | *Vulpes vulpes* | - | Y | Perforation | | |
| 16 | KEB 37.595 | Bone Bead | Phalange bead | *Gazella gazella* | - | Y | Lateral Right | | |

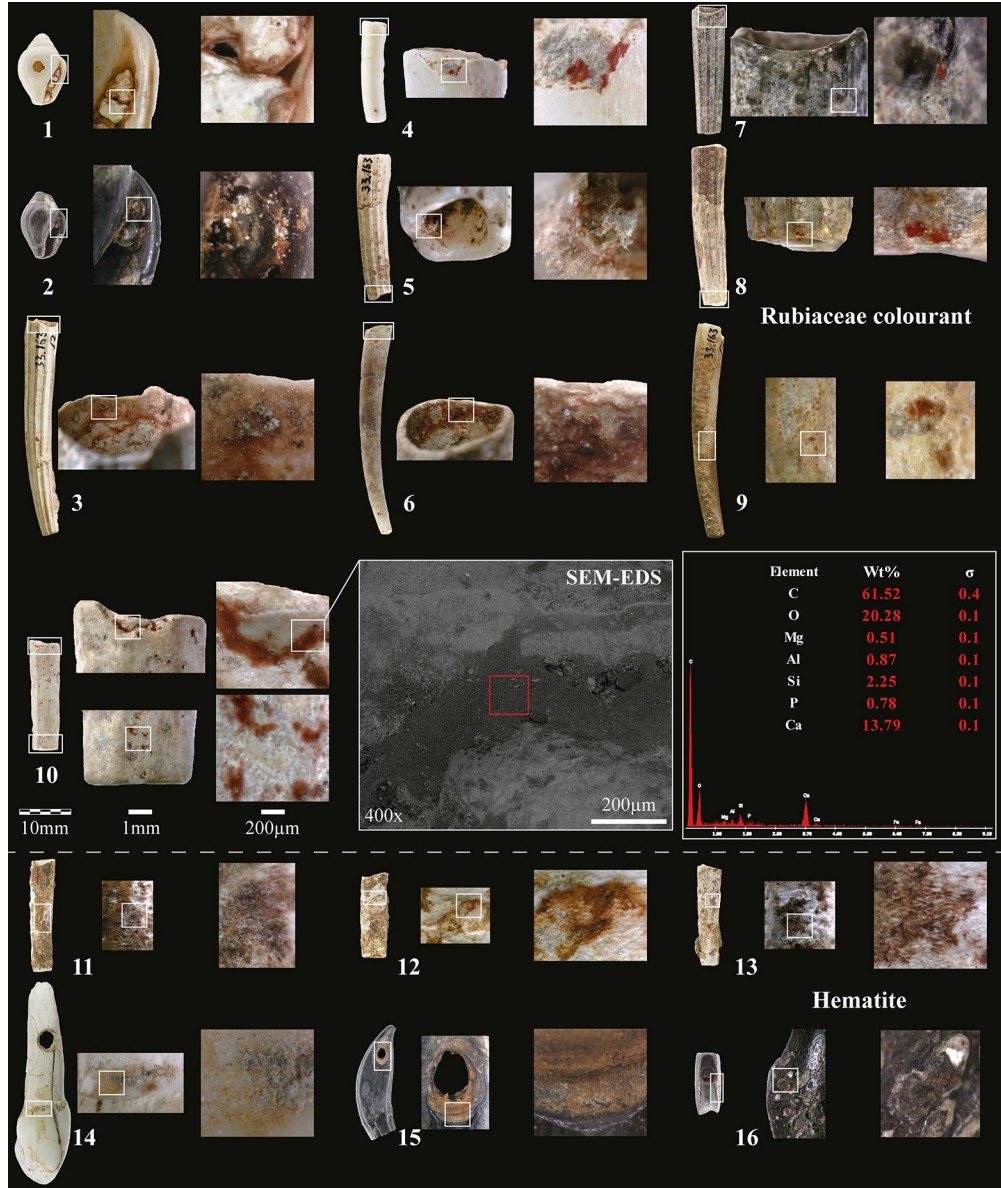

Element | Wt% | σ
--- | --- | ---
C | 61.52 | 0.4
O | 20.28 | 0.1
Mg | 0.51 | 0.1
Al | 0.87 | 0.1
Si | 2.25 | 0.1
P | 0.78 | 0.1
Ca | 13.79 | 0.1

**Fig 2.** Shell, tooth and bone beads and pendant with organic Rubiaceae colourant (1–10) and iron-oxide (11–16) coloured residues from the Early Natufian of Kebara Cave. 1–2: Columbella beads; 3–13: Scaphopod beads; 14: Large ungulate incisive pendant; 15: Fox canine pendant; 16: Gazelle phalange bead. N°10: *Dentalium sexangulum* with an SEM image in secondary electrons mode showing the location of the SEM-EDS analysis of its coloured residue. It indicates the absence of iron and a high carbon content, suggesting an organic compound (photos L.D.).

distribution of colouring on the artefacts (e.g., beads rubbing against coloured surfaces, beads intentionally coloured, threads holding the beads impregnated with a colourant [43]). However, the distribution of coloured residues in different parts of the beads suggests intentional colouration.

A group of six beads (Fig 2.11-2.16) showed residues identified as hematite (iron-oxide) by Raman spectroscopy (Fig 3). Blocks of hematite discovered at Kebara also attest to the intense use of this colourant by its Natufian occupants (Fig 4). The colour and consistency of the hematite on the personal adornments seem to vary according to the bead type maybe

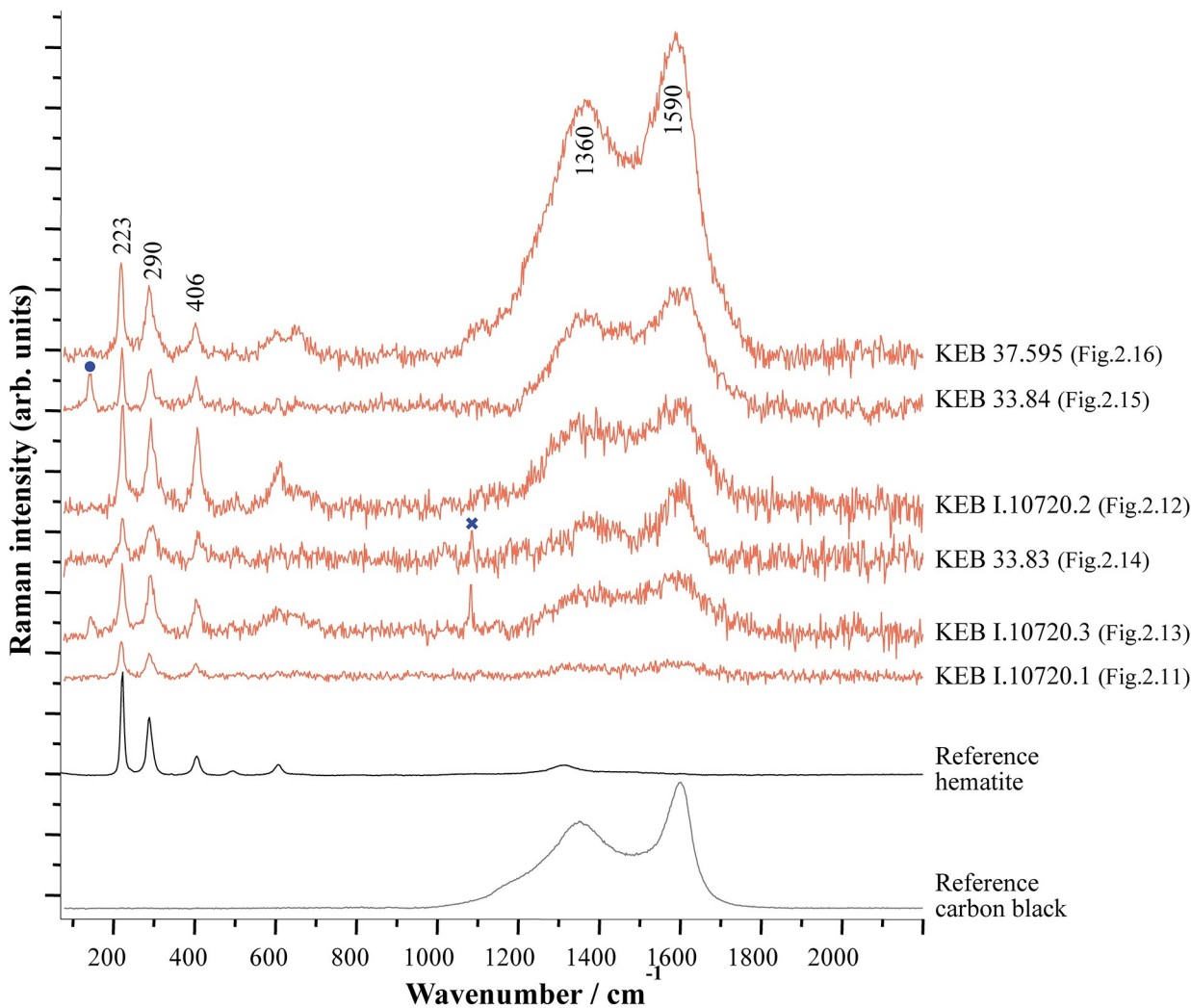

**Fig 3. Raman spectra (λexc = 458 nm) obtained on Kebara Cave Early Natufian beads and pendants suggesting the presence of hematite.**
Some spectra also show the presence of carbon black. The cross indicates the main band of calcite observed on some spectra obtained on shells, suggesting a contribution from the substrate. The dot matches the main anatase band due to titanium impurities in some iron oxides. For comparison, reference spectra of hematite and carbon black are also presented. The spectra were adjusted to show comparable maximum intensities.

portraying the variability of the hematite outcrop source or the transformation processes before use, or even taphonomic effects. Carbon black (produced, for example, by the incomplete combustion of vegetable matter) was identified as admixed with hematite in some analysed locations (Fig 3). However, it is impossible to prove an intentional mixing of hematite and carbon black based solely on these surface analyses. The carbon black is ubiquitous in archaeological sediments and surface contamination cannot be excluded. Only the identification of prepared colouring blocks (hematite mixed with carbon black) could prove such practices [44], so far not identified in a Natufian context.

Conversely, all the colour residues found on the remaining ten beads, all shells (Fig 2.1-2.10), are deep, brilliant red, and seem homogenous in their dense consistency (Fig 5). Scanning Electron Microscopy with Energy Dispersive X-ray Spectroscopy (SEM-EDS) analysis of one of these beads (Fig 2.10) shows the absence of iron (Fe) thus ruling out that the red pigment is

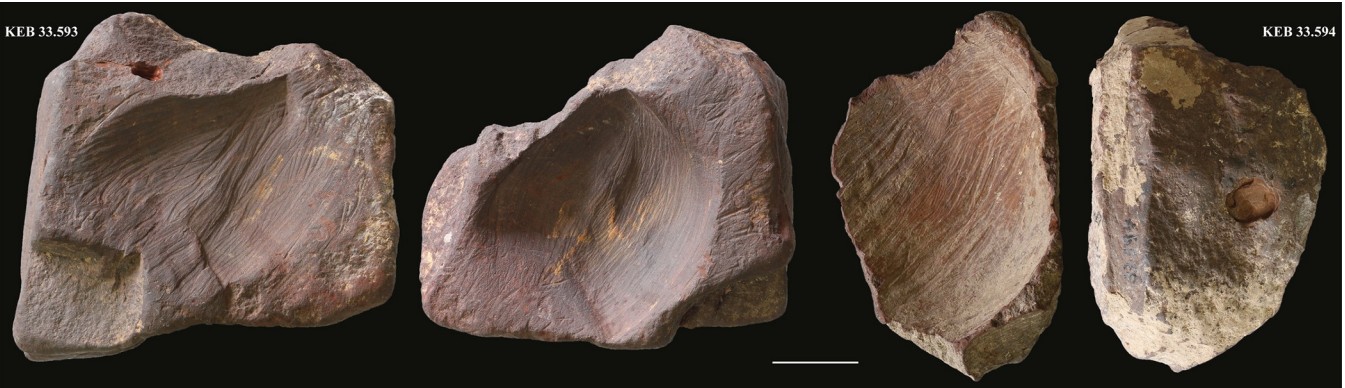

**Fig 4. Blocks of hematite (KEB 33.593 and 33.594) with flint scraping marks from the Early Natufian of Kebara Cave (scale 2 cm) (photos L.D.).**

made of iron oxide/ochre. Oxygen (O) content was almost constant throughout the analysed pigmented area ($20.2 \pm 2$ w%). The content of Carbon (C = 61,5 w%) is too high to be justified only by the presence of a calcareous concretion ($CaCO_3$) or by the calcium carbonate composition of the shell (Ca content is 13,8 w%). It is, therefore, necessary to invoke the presence of organic matter to explain the origin of the colour [9]. In order to identify it, we analysed the ten red-stained shell beads by Raman spectroscopy (Fig 6) (details in method section), a non-destructive technique broadly used to characterise organic (or inorganic) colouring matters. Surface Enhanced Raman Spectroscopy (SERS) or High-performance Liquid Chromatography (HPLC) could have been more specific identification techniques. Still, both are invasive and, for HPLC, require relatively large sampling, conditions of use not approved in the present study.

The spectra obtained from all the beads show very similar profiles with identical characteristic bands, of which some are slightly less visible due to the quality of the tenuous signals recorded (Fig 6). The spectrum is dominated by a massif between 1300 and 1700 $cm^{-1}$ characteristic of cyclic organic molecules [45] with different positions of characteristic bands shown in Fig 6. For these red dyes, it is, therefore, a question of specifying the relevant molecule of the chromophore family from animals or plants [46, 47]. In prehistoric Levant, several species of plants and animals could have been used by the Natufians to produce a red organic colourant

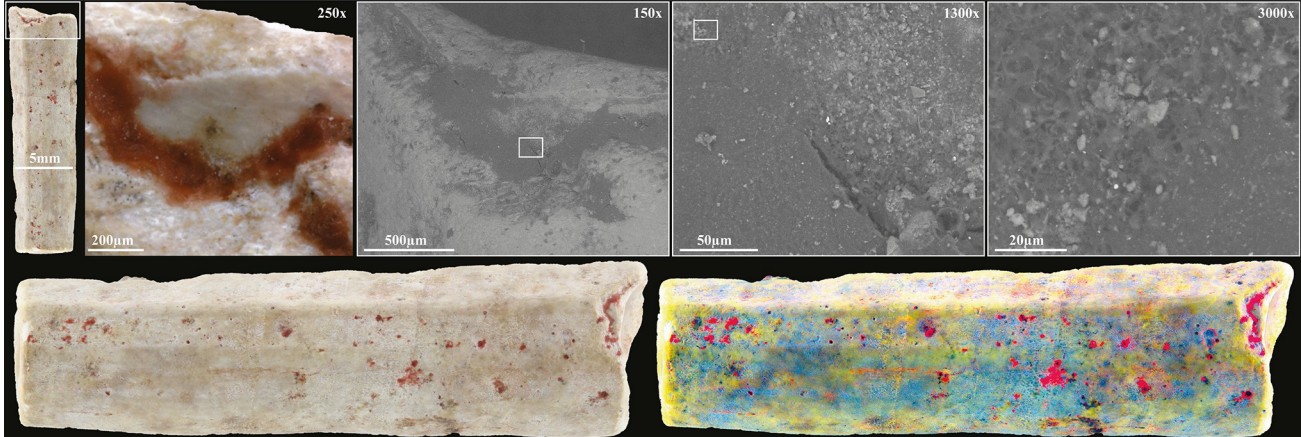

**Fig 5.** Detail of the organic colourant observed on a *Dentalium sexangulum* bead (KEB 337336.3: Fig 2.10) with SEM images at different magnifications and a DStrech view (at right), which makes it easier to see the distribution of organic colourant residues (in red) on the surface of the bead (photos L.D.).

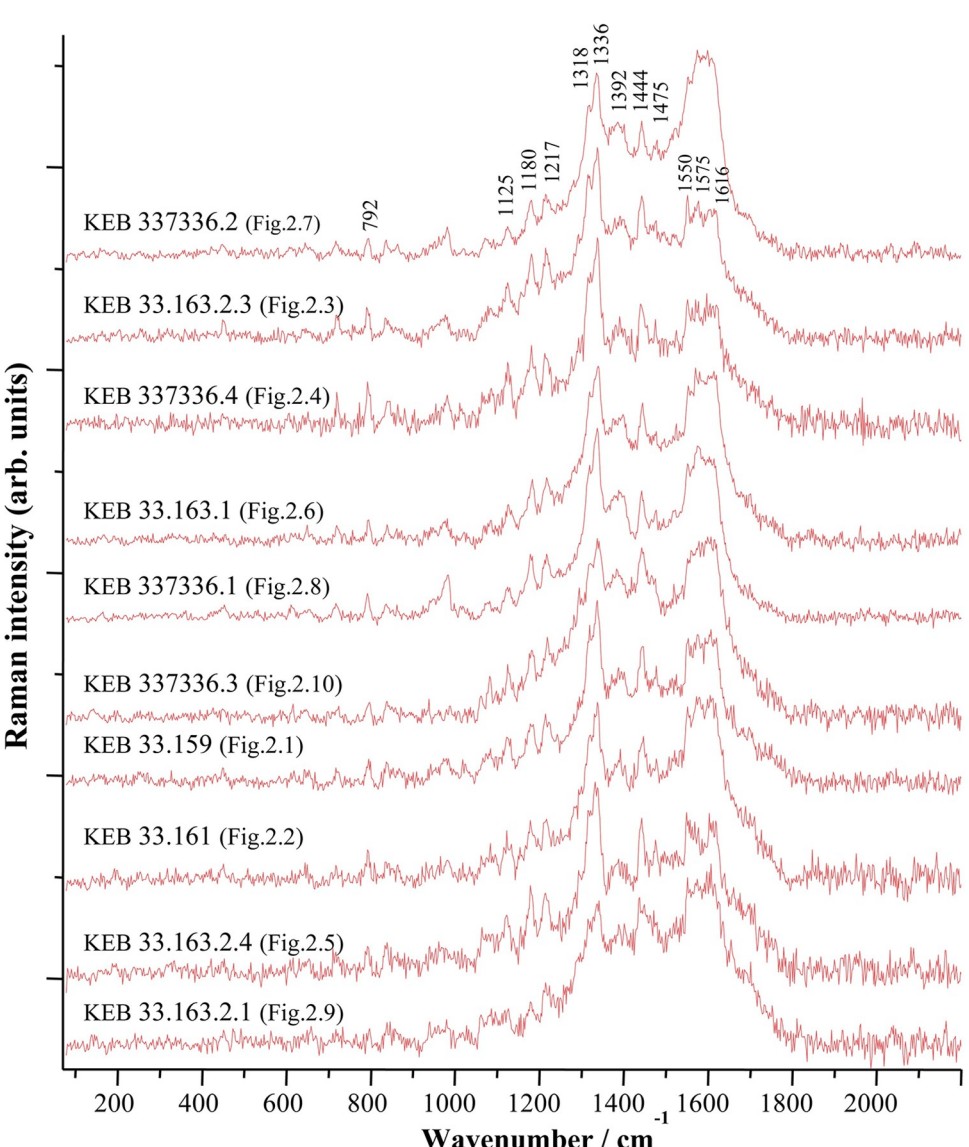

**Fig 6. Raman spectra (λexc = 458 nm) obtained on Kebara Cave Early Natufian beads, suggesting the presence of an organic dye.** The spectra were adjusted to show comparable maximum intensities.

[11] (Table 2). The Raman spectra of these dyes available in the literature are mainly spectra obtained using SERS. This technique results in substantial changes in the relative intensities of the different bands in the spectra, resulting in significantly modified spectral profiles. Therefore, the spectra obtained using SERS are not directly comparable with those obtained using conventional Raman and cannot be used for formal identification. Nevertheless, to examine whether this could explain the presence of specific bands in the spectra obtained on the Kebara organic dyes, we propose to compare some of them from the literature with our spectra in Fig 7 (as well as with conventional Raman spectra for two compounds). The comparison shows that the characteristic bands of the Natufian colourant are not fully represented in the reference Raman or SERS spectra of dye molecules spectra or plants as the Dyer's alkanet (Alkannin), Henna (Lawsone), Orchil (Orcein) or scale insects (Carminic acid or Kermesic acid) (Table 2).

**Table 2. List of the plant and animal species that could have been used to produce a red organic colourant in the prehistoric Levant [11].**

| Colourant source | | | | Part to use | Chromophore Type | Main Chromophores |
|---|---|---|---|---|---|---|
| **Type** | **Family** | **Common name** | **Latin name** | | | |
| Plants | Rubiaceae | Domesticated common madder | *Rubia tinctorum* | Root | Anthraquinones | Alizarin Purpurin Pseudopurpurin |
| | | Slender-leaved madder | *Rubia tenuifolia* | | | |
| | | Wild common madder | *Rubia peregrina* | | | |
| | | Dyer's woodruff | *Asperula tinctoria* | | | |
| | | *Asperula spp. Southern Levant* | *Asperula libanotica* | | | |
| | | | *Asperula arvensis* | | | |
| | | | *Asperula setosa* | | | |
| | | | *Asperula glomerata* | | | |
| | | Lady's bedstraw | *Galium verum* | | | |
| | | *Gallium spp. Mediterranean Basin* | *Galium saxatile* | | | |
| | | | *Galium odoratum* | | | |
| | | | Galium mollugo | | | |
| | Boraginaceae | Dyer's alkanet | *Alkanna tinctoria* | | Naftoquinones | Alkannin |
| | Lythraceae | Henna | *Lawsonia inermis* | Leaf | | Lawsone |
| | Roccellaceae | Orchil | *Roccella spp.* | Lichen | Orchil dyes | Orcein |
| Insects | Kermesidae | Vermilion | *Kermes vermilio* | Body of the female | Anthraquinones | Kermesic Acid |
| | Margarodidae | Carmine | *Porphyrophora spp.* | | | Carminic Acid |

The closest match (without being a definite identification) was found with signatures from plant compounds, the anthraquinones (alizarin: $C_{14}H_8O_4$ and purpurin: $C_{14}H_8O_5$) isolated from the inner root parts of the Rubiaceae family (*Rubia* spp., *Asperula* spp., *Gallium* spp. (Table 2)) [11]. The presence of Rubiaceae species in Mount Carmel at the end of the Pleistocene is attested in a pollen analysis from the Early Natufian layers of el-Wad Cave, a few kilometres north of Kebara cave [48] (Fig 1A and 1B). *Rubia* spp. (Madder), the most commonly used Rubiaceae species for colouring [11], has been used across the world by dyers [49] and painters [50] as a highly valued red pigment from the IV[th]/III[rd] millennium BCE in the Iberian Peninsula [9] and the Indus Valley [10] until the invention of synthetic alizarin by German chemists, Graebe and Liebermann, in 1868 [51]. For example, madder-dyed textiles and pigments have been identified in Tutankhamun's tomb [52], on the Shroud of Turin [53], and in the paintings of Van Gogh [54]. Today, this family of perennial plants is represented in the southern Levant and Mount Carmel by the domesticated common madder (*Rubia tinctorum* L.) [55] and the wild slender-leaved madder (*Rubia tenuifolia* L.) [56] with alizarin as the main anthraquinone isolated from their roots [57]. Wild common madder (*Rubia peregrina* L.), with purpurin as the main anthraquinone isolated from its roots [58], might have also been present in the southern Levant during earlier periods (before madder domestication whose origin has not yet been established) given that its habitat in the wild encompassed the whole Mediterranean basin up to the Caucasus [59, 60].

With some representative Raman spectra obtained on the Kebara beads, Fig 8 shows the Raman spectra obtained on the main anthraquinones of Rubiaceae (purified dye of alizarin and purpurin from Sigma-Aldrich and Sennelier) isolated from the roots of domesticated common madder (*Rubia tinctorum* L.), a modern commercial madder lake (mixture of alizarin and purpurin from Kremer Pigmente) and directly measured on the coloured parts of the roots and berries (unprocessed) of wild common madder (*Rubia peregrina* L.) that we gathered near Paris, slender-leaved madder (*Rubia tenuifolia* L.) that we gathered near Jerusalem (S1 Fig) and domesticated common madder (*Rubia tinctorum* L.) bought from a herbalist. The

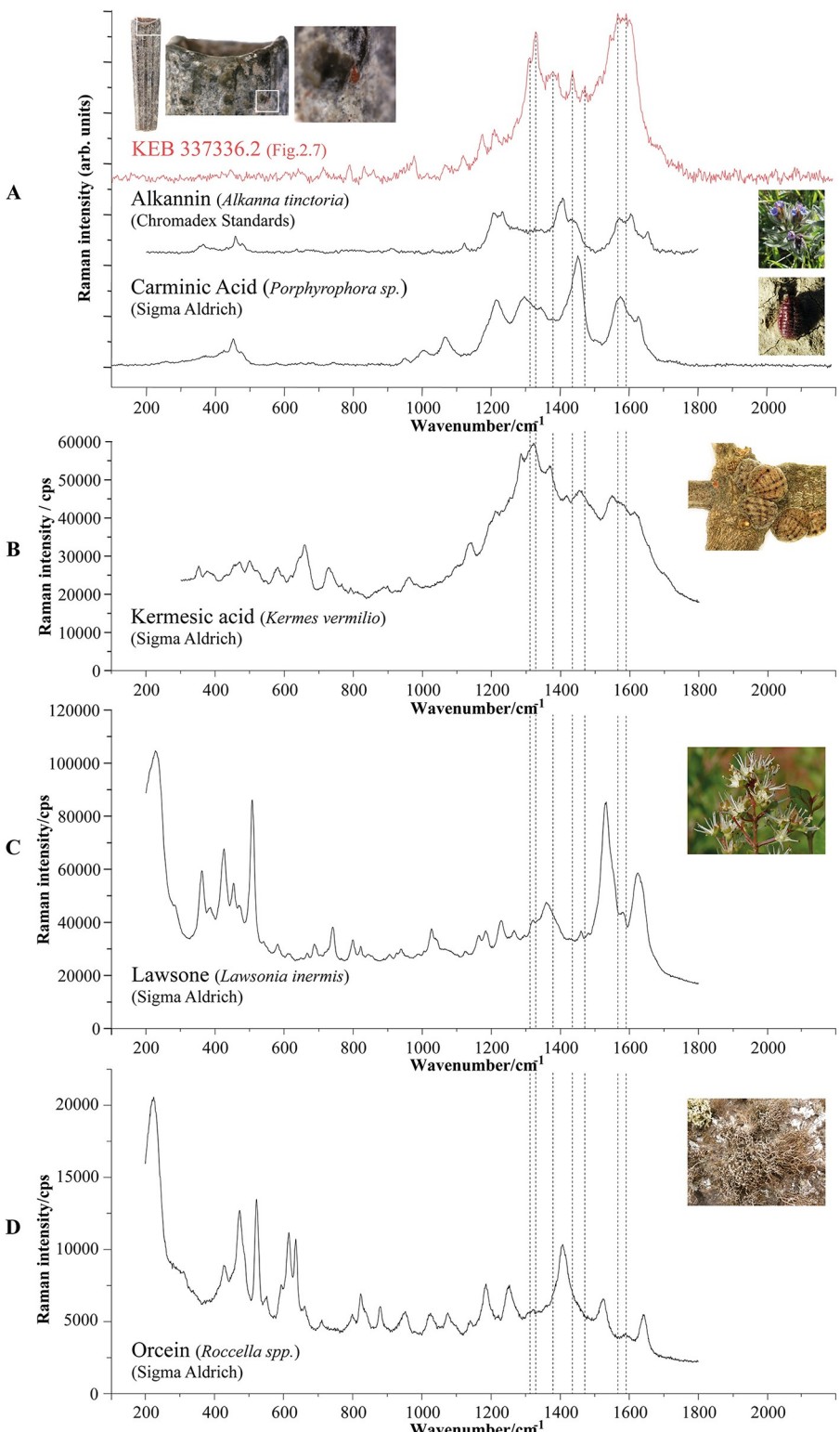

**Fig 7.** (A). Raman spectra obtained on an Early Natufian Kebara Cave scaphopod bead (in red) compared to Raman spectra (λexc = 448 nm) from Alkannin (Chromadex standards after [61]); Carminic acid (Sigma Aldrich measured in this study λexc = 458 nm); (B). SERS spectra (λexc = 785 nm) of Kermesic acid (extract of kermes insects after [62]); (C) SERS spectra (λexc = 785 nm) of Lawsone (Sigma Aldrich after [62]); (D) SERS spectra (λexc = 785 nm) of Orcein (Sigma Aldrich after [62]). Vertical dashed lines are drawn in correspondence with characteristic archaeological bands.

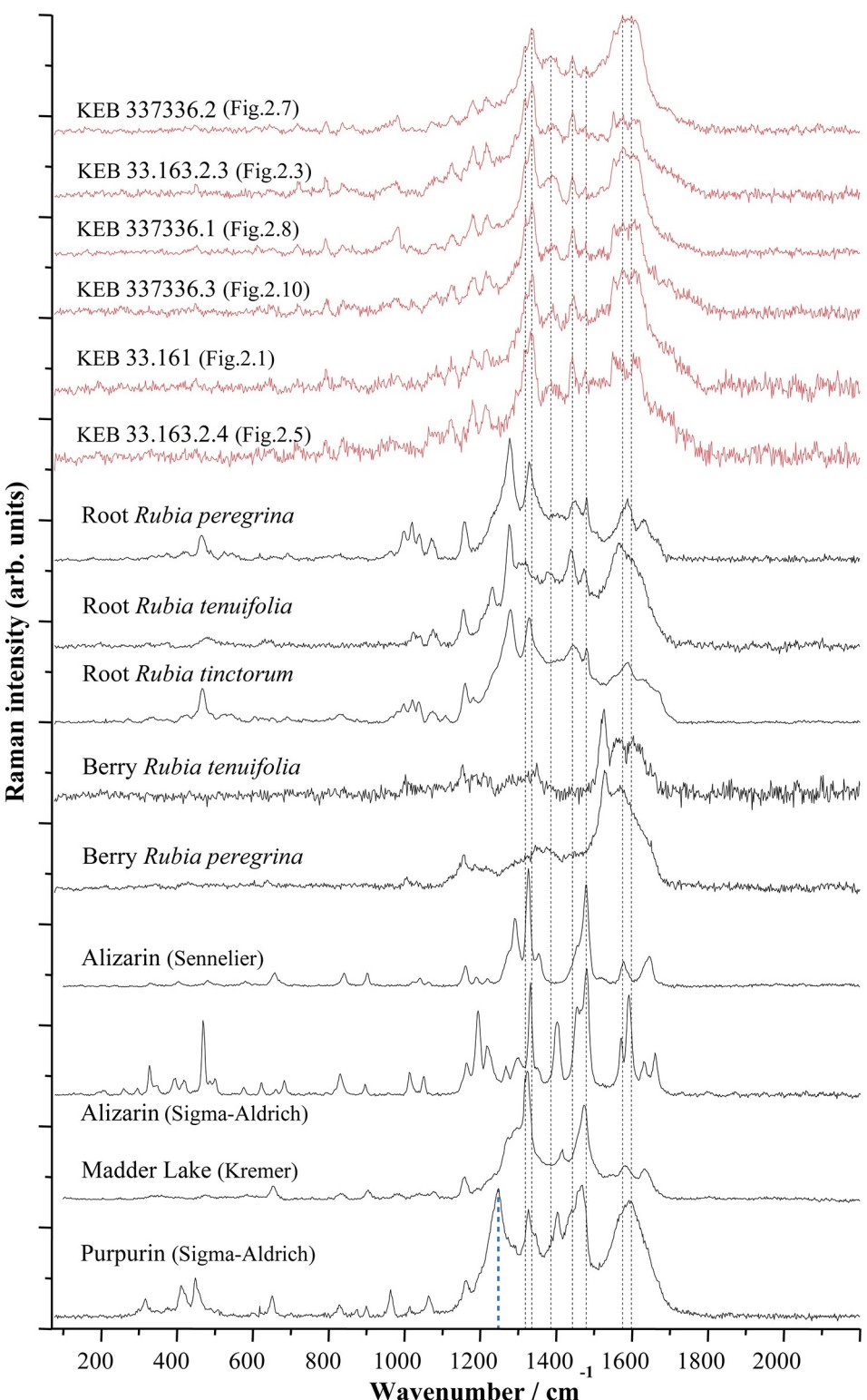

**Fig 8. Selected Raman spectra (λexc = 458 nm) obtained on Early Natufian Kebara Cave beads (in red) evidencing the presence of an organic dye compared to spectra from various origins (plants, commercial dyes and pure molecule).** A baseline was removed from all spectra to avoid fluorescence backgrounds and the spectra were adjusted to show comparable maximum intensities. Vertical dashed lines are drawn in correspondence with characteristic bands from Kerara artefacts (see also Fig 9 for a comparison of some of these spectra).

Raman spectra of these products are different because of their nature, *Rubia* spp. roots being unprocessed and commercial references (Sennelier, Sigma-Aldrich and Kremer Pigmente (Fig 8)) resulting from a process of extraction/purification or synthesis that, obviously, would not have been possible in Prehistory. Similarities can be pointed out between Natufian colourant and modern references, even if there is no perfect match. Although the spectra obtained do not allow the origin of the Natufian colourant to be identified with certainty, everything suggests that it belongs to the family of anthraquinones derived from Rubiaceae plants. Bearing in mind that natural plant substances are complex mixtures of many molecules, differences could be explained by the characteristics of the Rubiaceae species that were accessible to the Natufians, other anthraquinone molecules such as pseudopurpurin, xanthopurpurin, rubiadin and munjistin could also be present in the Natufian colourant, the relative amounts of which vary with the age of the plant [63].

Concerning the madder, even when using a specific plant species, the final composition of a madder colourant can be significantly affected by several factors such as cultivation parameters, harvesting and storage conditions of the roots [64–66], effects of the colourant extraction process (e.g., drying time, heating temperature, fermentation intensity) [60] and the method used for colourant preparation [64, 67, 68]. In addition to the chemistry of the original products, time and the conservation environment may have modified the dye molecules. The taphonomic effects of the acidic conditions of Kebara cave sediment might have transformed the initial colourant (even if it preserved some colour and a molecular signature), knowing that Raman spectra of dyes are strongly influenced by Ph [60, 69, 70]. Furthermore, up to date, there is no other Raman reference for such an old complex organic dye that would allow us to make a comparison in a similar state of temporal degradation.

For example, the Raman spectra of reference purpurin from Sigma-Aldrich exhibits a strong signal at 1256 cm$^{-1}$ (blue vertical dashed lines at the bottom of Fig 8), not shown on the spectra from Kebara beads, which is characteristic of pseudopurpurin [71], an unstable anthraquinone that covert to purpurin over time (decarboxylation: loss of carbon dioxide) [71] or depending on soaking and drying conditions of the roots [60, 72]. Also, despite being the closest match to the Natufian colourant (Fig 9), the Raman spectra from unprocessed roots of wild common madder (*Rubia peregrina*), slender-leaved madder (*Rubia tenuifolia*) and domesticated common madder (*Rubia tinctorum*) exhibit a strong signal at ca. 1290 cm$^{-1}$ not shown in the Kebara spectra (Fig 9). This strong signal is not shown on the reference Madder Lake (a mixture of purpurin and alizarin made of *R. tinctorum* roots) (Fig 9). It, therefore, seems to correspond to transformation occurring during the many stages of transformation from fresh root to finished colourant, an observation confirmed by experimental manufacture of alizarin and purpurin lake [46, 72]. A direct comparison of the Raman spectra of the Natufian colourant with reference molecules of Purpurin and Alizarin from Sigma Aldrich (Fig 10) shows that many of the bands characteristic of this dye are also found in the spectra of these two main anthraquinones molecules originating from the Rubiaceae roots (Table 2). The mismatch in relative intensities, which is an essential feature of Raman signatures, does not allow formal identification. Still, these comparisons suggest the similarities that we have been able to underline between the Natufian organic colourant and compounds from Rubiaceae plants.

## Discussion and conclusion

There is no evidence that these organic colour residues may result from a recent contamination of Kebara Cave Natufian beads. Rubiaceae natural habitat is the Mediterranean maquis and forest [11]. Therefore, they do not grow inside caves, thus ruling out the possibility of a natural deposition from the plant's roots into the archaeological layer. Furthermore, given the

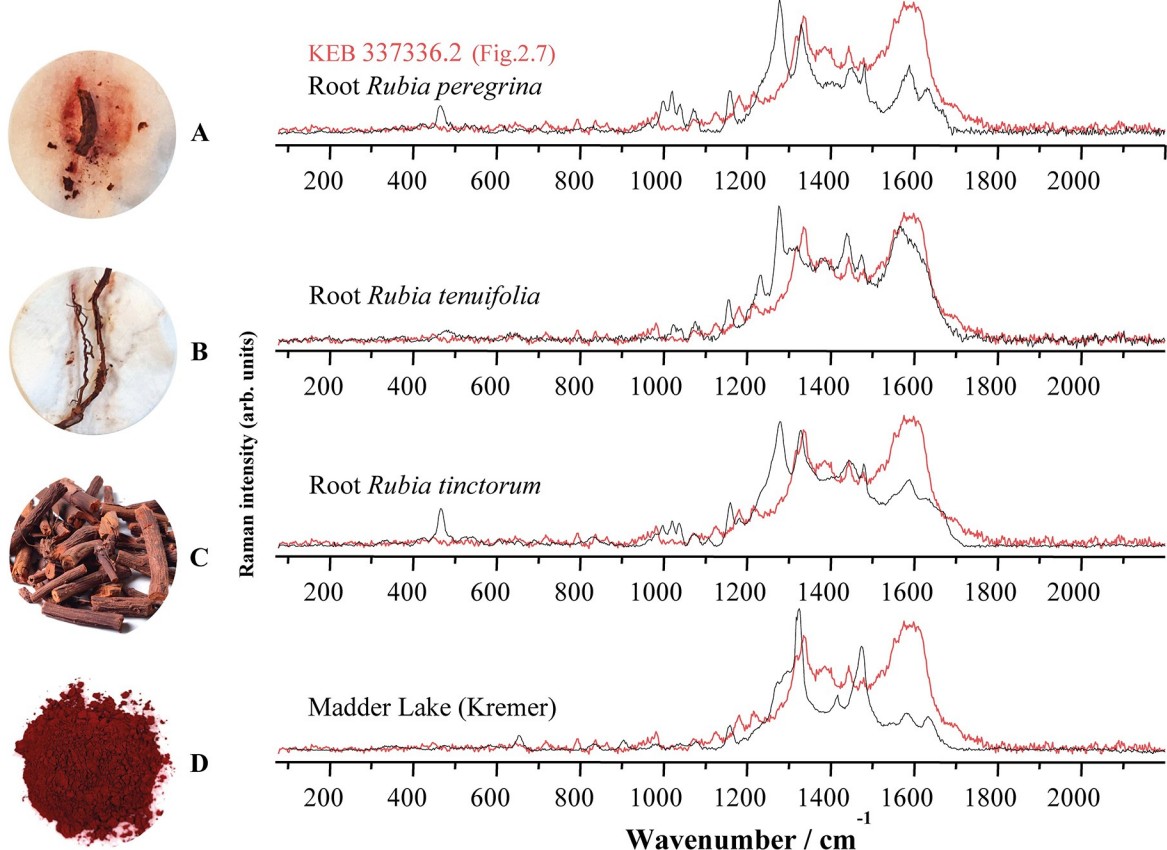

**Fig 9.** Raman spectra (from Fig 8) obtained on an Early Natufian Kebara Cave scaphopod bead (in red) superimposed on the spectra from the unprocessed roots of (A) wild common madder (*Rubia peregrina*), (B) slender-leaved madder (*Rubia tenuifolia*), (C) domesticated common madder (*Rubia tinctorum*) and (D) a commercial Madder Lake from Kremer Pigmente. A baseline was removed from all spectra to avoid fluorescence backgrounds and the spectra were adjusted to show comparable maximum intensities (photos L.D.).

high density of artefacts in the Natufian occupation layers, it seems unlikely that only the shell beads would be affected in case of contamination. Still, we have observed these residues only on this type of adornment. Contamination after the excavation in the Rockefeller Museum in Jerusalem seems unlikely given that the same type of deep brilliant red colour residues, yet unstudied, can be observed on Early Natufian scaphopod beads from Kebara Cave curated at the Peabody Museum of Harvard University since 1934 (S2 Fig). Above all, the distribution of the organic (Rubiaceae) and mineral (ochre) residues on all the beads is similar and sometimes covered by post-depositional concretion (Fig 11), thus ruling out the possibility of post-depositional contamination. Similarly, the attribution of these beads to the Early Natufian (Layer B) does not seem to be debatable given that this layer is separated from the previous one (Layer C: Kebaran) by a sterile layer (Fig 1D) and that, typologically, the shell species used are typical of the Natufian culture and differ from what can be observed in the Neolithic and later cultures found in the succeeding Layer A above [73, 74].

Contamination being excluded, the Early Natufian beads from Kebara Cave, older by 9,000 years from the first case of Rubiaceae use as a pigment published so far [9, 10], are thus the earliest reliable evidence of the manufacture and use of organic red pigments. It demonstrates genuine innovation in ornamental practices and the *chaîne opératoire* of pigmenting materials at the beginning of the sedentary lifestyle. The gathering, processing and use of the

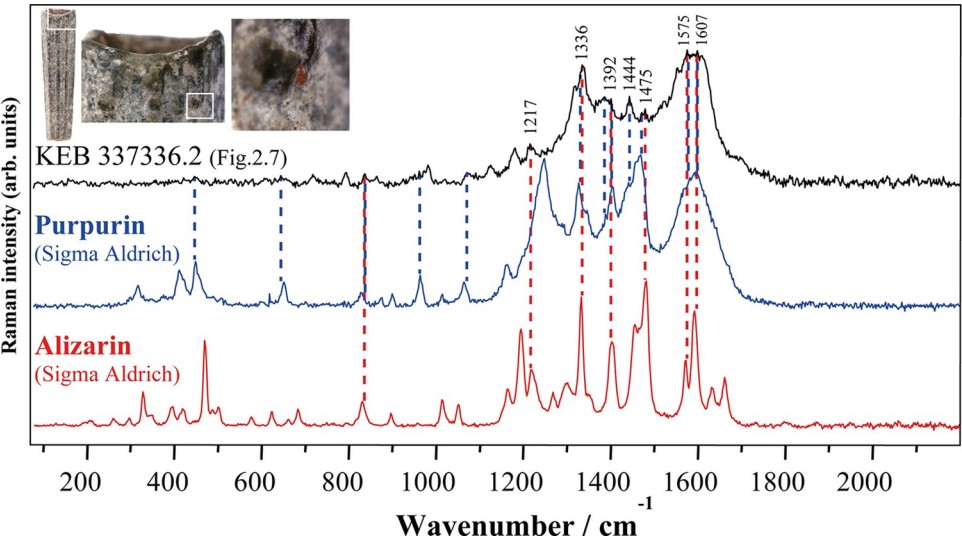

**Fig 10. Raman spectra (from Fig 8) obtained from an Early Natufian Kebara Cave scaphopod bead (in black) compared to spectra of purpurin and alizarin molecules from Sigma Aldrich.** Vertical dashed lines are drawn in correspondence with characteristic Kebara dye bands (photos L.D.).

underground part of a non-dietary plant by the Natufians demonstrate the development of apparent botanical knowledge that had not been suggested previously in the southern Levant.

We do not know the ways employed by the Natufians to transform the roots of Rubiaceae into colouring matter, given that the red colour residues that we observed on some of the stone tools from Kebara (Fig 12) have yet to be studied in detail. On the other hand, we may assume that the manufacturing process of these plants mainly goes through the same stages in various cultures of historical periods [11]. Concerning *Rubia* spp., the gatherer dug to extract the Madder roots, probably having chosen the largest plants with the largest roots where the anthraquinonoid colourants are the most concentrated [75]. The roots are then cleaned, dried, hulled, crushed and boiled in hot water to dissolve the dye and fermented to hydrolyse

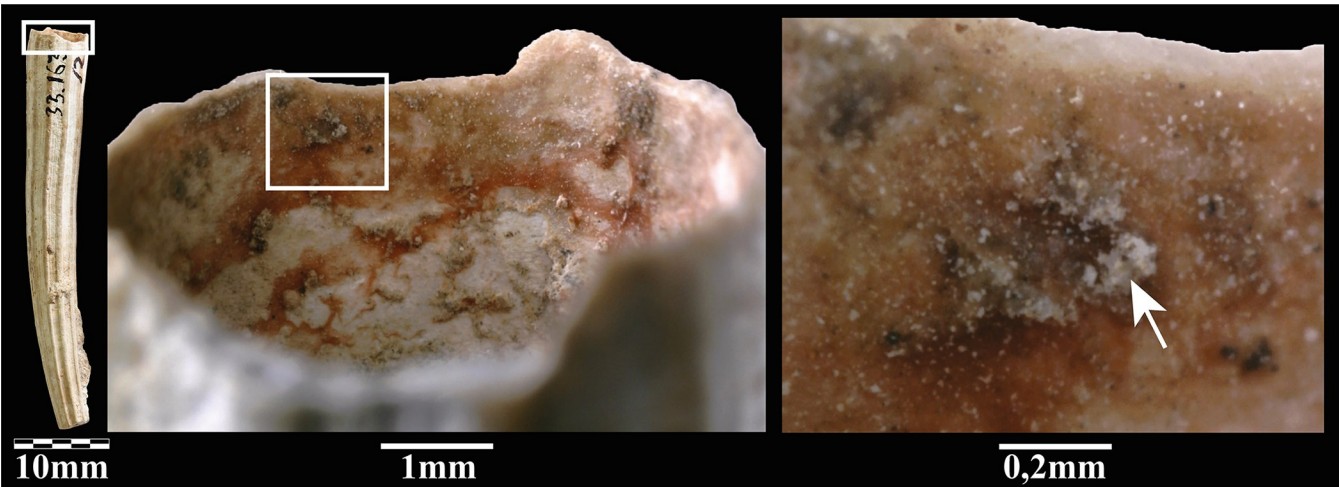

**Fig 11. Detail of Rubiaceae colourant residue inside a scaphopod bead (KEB 33.163–2) from the Early Natufian layer of Kebara Cave.** Note that the colour residue is covered by a post-depositional concretion (white arrow), thus ruling out the possibility of a recent contamination. Magnification 50x and 250x (photos L.D.).

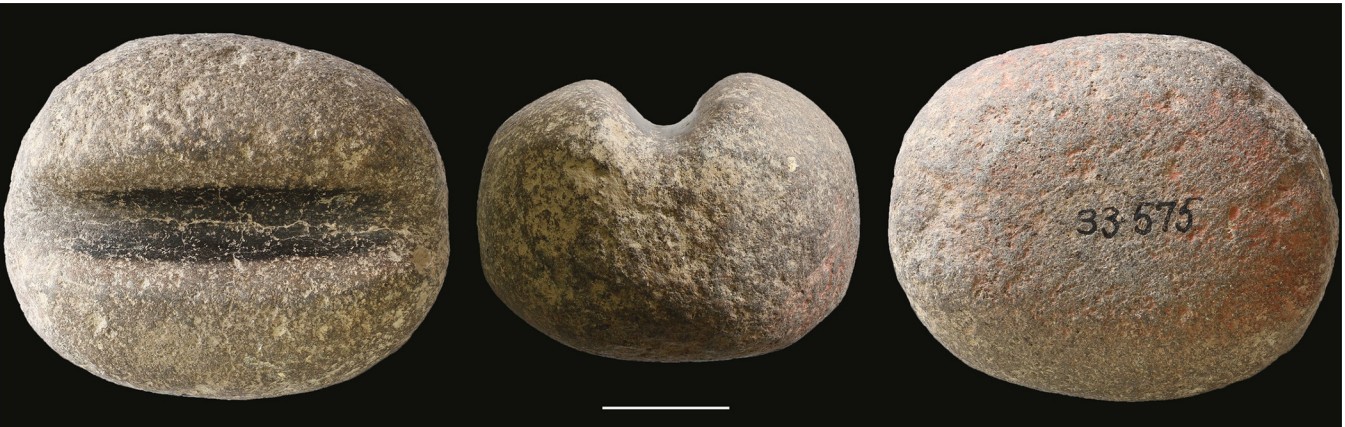

**Fig 12. A grooved stone tool (KEB 33.575) with red colour residues (on its lower surface) from the Early Natufian of Kebara Cave (scale 2 cm) (photos L. D.).**

anthraquinones from the glycosides [9]. The SEM-EDS (Fig 2.10) is unclear whether the Natufians used mordant to fix the anthraquinones, as observed in later periods for textile dyeing [11]. Future studies of experimental manufacture of pigments from various Rubiaceae species coupled with invasive analytical methods on the archaeological beads and tools from Kebara could help identify the actual plant species used by the Natufians and its *chaîne opératoire*. In any case, it seems more extended and more complex than the *chaîne opératoire* of mineral pigments, such as that of ochre widely used by the Natufians [15–24]. Therefore, one might ask why they manufactured and used organic red dyes if they already used mineral red pigments? The answer may lie in the intensity of the Rubiaceae red, which, as we have seen (Figs 2 and 5), is more intense and has more tinting power than the ochre red. If confirmed, the mixture of organic carbon black (produced, for example, by the combustion of vegetable matter) with the mineral ochre (which increases the darkness of the red) observed on some Kebara beads (Fig 3) could correspond to the same objective of looking for the reddest, most chromatic pigment. Such behaviours are known in other Early Natufian sites as el-Wad cave, to the north in Mount Carmel (Fig 1A and 1B), where red hematite has been produced by heating locally available yellow goethite [76, 77], or Hayonim Terrace, in western Galilee (Fig 1A and 1B), where a well-mastered and controlled use of fire was practised by skilled craftsmen to colour in red the locally available iron-rich yellow chert [78].

The search for a chromatic variation of red, through a significant technical and time-consuming investment, echoes other practices typical of Natufian ornamental traditions. This can be seen, for example, in Eynan-Mallaha, Hayonim and el-Wad (Fig 1A and 1B) during the Early Natufian with the different species of Mediterranean scaphopods transformed into beads that are, morphometrically and visually, quite similar, yet are collected, transformed and used differently within an individual's adornment [21, 22]. In Kebara cave, the sophisticated colouration of the oval bone pendants by controlled heating [39], producing a rich variation of shades from white, grey, brown to black (Fig 13), indicates the same type of behaviour. This leads to the conclusion that, in their ornamental practices, the Natufians strived to invest meaning and convey a different message through barely perceptible variations such as different chromatic variations of the red colour. Ethnographic accounts illustrating such behaviours are known from the San hunter-gatherers in southern Africa, who distinguished and used specific shades of red ochre for various body decorations [79], or the Ovahimba hunter-pastoralists from Namibia, who use a particular shade of red ochre as a body cosmetic [80]. This

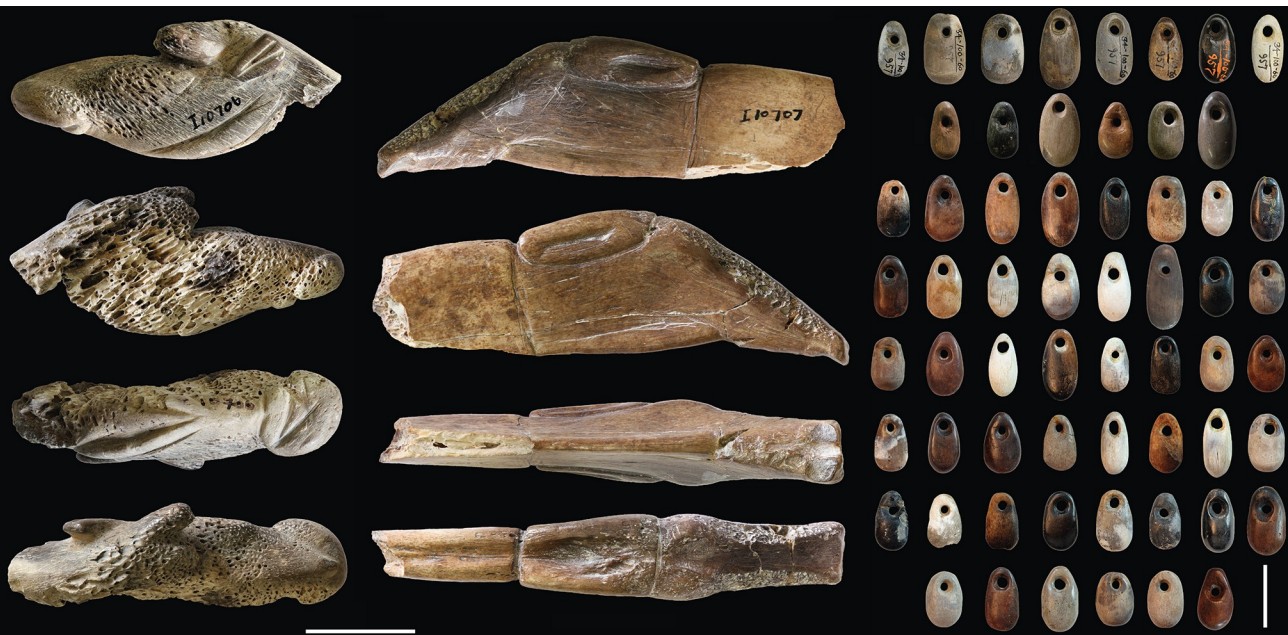

**Fig 13. Selected bone zoomorphic figurations and oval bone pendants from the Early Natufian of Kebara Cave (scale 2 cm) (photos L.D.).**

behaviour of strong symbolic investment in a specific shade is also attested in Historical periods with, for example, Tyrian purple. This dye, isolated from Muricidae marine gastropods by a considerable investment, was reserved, under penalty of death, to be worn by Roman emperors exclusively [11]. The differences in Natufian red shades might have played an essential role in the processes of information exchange about individuals and the expression of personal and group identities [81] that increased with the advance of sedentism and territoriality during the late Epipaleolithic [14].

Apart from Kebara Cave findings, we have observed many mineral colour residues [21] among the more than ca. 10,000 bone, tooth and shell beads and pendants that we have studied at other Natufian sites in the southern Levant, spanning the whole 3,000-year duration of this culture (Hayonim, el-Wad, Eynan-Mallaha, Erq el-Ahmar), yet, we have never observed a red colouring material with the dense and bright appearance corresponding to the organic compound detailed herein here (LD, personal observation). These results also underline the contribution of analytical characterisations to identify and describe the nature of the colouring materials and, more generally, to study their acquisition, production, application and use. Is the presence of this organic dye in Kebara linked to the good preservation conditions in the cave or to a strictly local technical innovation? Was this organic colourant reserved exclusively for shell beads? Only the study of a more significant number of Natufian personal adornments from other sites of the Levant, as well as the study of the collections from Kebara stored in the United Kingdom and North American museums, will enable us to answer such questions. In any case, this unique discovery opens new research perspectives. Can these residues preserving organic matter be dated by radiocarbon? Could a genomic analysis of Natufian Rubiaceae shed light on the history of Madder domestication? On a different aspect, the aerial parts of the wild madder, stems and leaves, have long been and are still used today in the traditional medicine of many cultures [11, 59, 82, 83] for their aphrodisiac, antibacterial, anti-oxidant, iron-rich properties, demonstrated through recent pharmaceutical studies [84]. If indeed we will be able to expand the research in an endeavour to recover such findings through the study of other

artefacts (e.g., groundstone vessels and tools), then we will enrich our knowledge of the Natufian culture and its practices in new, currently understudied, domains of existence.

## Materials and methods

### Archaeological methods

Excavations at Kebara Cave were conducted by F. Turville-Petre and C.A. Baynes, on behalf of the British School of Archaeology in Jerusalem and the American School of Prehistoric Research.

This rapid three-month excavation, during the spring of 1931, extended over the entire surface of the cave, amounting to ca. 300 m$^2$ with an average depth of about 3 m (900 m$^3$) without sieving [25, 27, 31]. The artefacts discovered were attributed to a layer; their exact position in this layer was not recorded.

### Kebara beads analyses

The collection of 79 beads and pendants (58 bones, 19 shells and 2 teeth) is curated at the Rockefeller Archaeological Museum in Jerusalem. Its typological composition doesn't seem biased given that the same typological variation can be seen in the collections curated in museums in the United Kingdom and North America. The 16 beads were analysed microscopically with a stereoscopic Olympus SZX10 microscope (magnification 7,8–78 ×) and a Dino-Lite AD-7013MZT digital microscope (magnification 30–250 ×) at the *Centre de Recherche Français à Jérusalem*. Pictures were taken with a Canon 7D camera (180 mm Tamron Macro lens) with Helicon Remote, Helicon Focus and DStrech softwares.

### SEM-EDS

Scanning Electron Microscopy with Energy Dispersive X-ray Spectroscopy (SEM-EDS) is a non-destructive analytical method used for the elemental analysis or chemical characterisation of a sample [85]. It relies on an interaction of some source of X-ray excitation and a sample. SEM-EDS microanalyses were performed at the Centre for Nanoscience and Nanotechnology (Unit for Nano Characterization) of the Hebrew University of Jerusalem on a FEI Quanta 200 ESEM equipped with an EDAX EDS detector. We used the Low-Vacuum mode with 0.38 Torr chamber pressure and an accelerating voltage of 15–20 kV.

### Raman spectroscopy

Raman spectroscopy is a non-destructive method for observing and characterising a material's molecular composition and external structure, exploiting the physical phenomenon whereby a medium slightly alters the frequency of the light travelling through it [86]. This frequency shift, known as the Raman effect, corresponds to an energy exchange between the light beam and the medium, providing information about the substrate. Raman spectroscopy involves sending monochromatic light onto the sample and analysing the scattered light. The information obtained by measuring and analysing this shift can be used to determine specific properties of the medium using spectroscopy. Raman micro-analyses were performed in the MONARIS laboratory in Paris on a Horiba Jobin-Yvon HR800 dispersive spectrometer using the 458 nm radiation of an ionised Argon Laser (Coherent I-90C-6). Rayleigh filtering was achieved by an Edge filter, and the signal was analysed with 600 lines/mm grating and recorded by a Peltier-cooled CCD detector. The whole spectral window of interest was obtained in a single acquisition and a spectral resolution of about 4 cm$^{-1}$. Long working distance 50x and 100x microscope objectives were used to adjust the analysed volume to the

sample characteristics in a confocal configuration. These give a beam waist of about 3 and 1μm, respectively. To preserve sample integrity, the Laser power was adjusted to less than 300 μW on the sample. Acquisition time on artefacts was adjusted according to the sample and the point of analysis and was typically 5 to 45s repeated between 5 to 30 times. The presented spectra were corrected from the Edge filter contribution and baseline-corrected to remove the intense fluorescence contribution exhibited by the samples.

## Supporting information

**S1 Fig. Slender-leaved madder (*Rubia tenuifolia* L.) gathered near Jerusalem in November (photos L.D.).**
(TIF)

**S2 Fig. Deep brilliant red colour residues observed on Early Natufian scaphopod beads from Kebara Cave and curated at the Peabody Museum of Harvard University since 1934 (photos L.D.).**
(TIF)

## Acknowledgments

The authors wish to thank the following people for their help: the *Centre de Recherche Français à Jérusalem*, the Hebrew University of Jerusalem, the Fyssen Foundation and the Irene Levi Sala CARE Archaeological Foundation. Natalia Gubenko and Alegre Savariego from the Israel Antiquities Authority (IAA), Fawzi Ibrahim from the Rockefeller Archaeological Museum and Israel Museum. Susan Haskell and Olivia Herschensohn from the Peabody Museum of Harvard University. Thanks are also due to Sylvain Bauvais, who helped to gather slender-leaved madder in Jerusalem and Vitaly Gutkin from the Unit for Nano Characterization of the Hebrew University of Jerusalem. The authors are grateful for the valuable comments and suggestions provided by Anna Belfer-Cohen, Mina Weinstein-Evron and Elisabeth Davin-Mortier.

## Author Contributions

**Conceptualization:** Laurent Davin.

**Data curation:** Laurent Davin, Julien Navas.

**Formal analysis:** Laurent Davin, Ludovic Bellot-Gurlet.

**Funding acquisition:** Laurent Davin.

**Investigation:** Laurent Davin, Ludovic Bellot-Gurlet.

**Methodology:** Laurent Davin, Ludovic Bellot-Gurlet.

**Project administration:** Laurent Davin.

**Resources:** Laurent Davin, Ludovic Bellot-Gurlet, Julien Navas.

**Supervision:** Laurent Davin, Ludovic Bellot-Gurlet.

**Validation:** Laurent Davin, Ludovic Bellot-Gurlet.

**Visualization:** Laurent Davin, Ludovic Bellot-Gurlet, Julien Navas.

**Writing – original draft:** Laurent Davin, Ludovic Bellot-Gurlet.

**Writing – review & editing:** Laurent Davin, Ludovic Bellot-Gurlet.

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
