## [Decision Letter · Decision Letter 0]

24 May 2023

PONE-D-23-12821Plant-based red colouration of shell beads 15,000 years ago in Kebara Cave, Mount Carmel (Israel)PLOS ONE

Dear Dr. Davin,

Thank you for submitting your manuscript to PLOS ONE. After careful consideration, we feel that it has merit but does not fully meet PLOS ONE’s publication criteria as it currently stands. Therefore, we invite you to submit a revised version of the manuscript that addresses the points raised during the review process.

This is an interesting study; however, both the reviewers expressed concerns regarding the identification of the materials purportedly used.  For this contribution to be considered for publication, you will need to provide a full methodology, and all the relevant data, on an experimental programme covering a wide range of different materials and plants. You will then need to demonstrate, in detail, how you have reached your interpretations on the identification of the materials. The reader will need to be fully confident that they understand how you reached your results, and that, if required, this study is reproducible. 

We look forward to receiving your revised manuscript.

Kind regards,

Karen Hardy

Academic Editor

PLOS ONE

Journal Requirements:

2. In your manuscript, please provide additional information regarding the specimens used in your study. Ensure that you have reported human remain specimen numbers and complete repository information, including museum name and geographic location. 

For more information on PLOS ONE's requirements for paleontology and archeology research, see https://journals.plos.org/plosone/s/submission-guidelines#loc-paleontology-and-archaeology-research.

"The authors wish to thank the following people for their help: the Centre de Recherche Français à Jérusalem, the Hebrew University of Jerusalem, the Fyssen Foundation and the Irene Levi Sala CARE Archaeological Foundation who supported our work. Natalia Gubenko and Alegre Savariego from the Israel Antiquities Authority (IAA), Fawzi Ibrahim from the Rockefeller Archaeological Museum and Israel Museum. Susan Haskell and Olivia Herschensohn from the Peabody Museum of Harvard University. Thanks are also due to Sylvain Bauvais who helped to gather slender-leaved madder in Jerusalem and Vitaly Gutkin from the Unit for Nano Characterization of the Hebrew University of Jerusalem. The authors are grateful for useful comments and suggestions provided by Anna Belfer-Cohen and Mina Weinstein-Evron."

"Fyssen Foundation post-doctoral fellowship (LD)

https://www.fondationfyssen.fr/en/

Irene Levi Sala CARE Archaeological Foundation (LD)

https://www.prehistory.org.il/irene-levi-sala-care-archaeological-foundation-grants-in-aid/

Centre de Recherche Français à Jérusalem (LD)

https://www.crfj.org/en/

6. We note that Figure 1 in your submission contain [map/satellite] images which may be copyrighted. All PLOS content is published under the Creative Commons Attribution License (CC BY 4.0), which means that the manuscript, images, and Supporting Information files will be freely available online, and any third party is permitted to access, download, copy, distribute, and use these materials in any way, even commercially, with proper attribution. For these reasons, we cannot publish previously copyrighted maps or satellite images created using proprietary data, such as Google software (Google Maps, Street View, and Earth). For more information, see our copyright guidelines: http://journals.plos.org/plosone/s/licenses-and-copyright.

Reviewers' comments:

Reviewer's Responses to Questions

**Comments to the Author**

1. Is the manuscript technically sound, and do the data support the conclusions?

Reviewer #1: Partly

Reviewer #2: Yes

2. Has the statistical analysis been performed appropriately and rigorously? 

Reviewer #1: N/A

Reviewer #2: N/A

3. Have the authors made all data underlying the findings in their manuscript fully available?

Reviewer #1: Yes

Reviewer #2: Yes

4. Is the manuscript presented in an intelligible fashion and written in standard English?

Reviewer #1: No

Reviewer #2: Yes

5. Review Comments to the Author

Reviewer #1: This manuscript presents the discovery of an organic plant-based dye on shell beads dating to about 13,000 years cal BP. This discovery is unique and predates the previously known evidence of plant-based dyes from 7,000 years. The data are well presented and the methodology is in adequation with the goals of the paper. However, if the presence of on organic dye is demonstrated, the interpretation that it comes from madder roots would benefit of further discussion. Indeed, it is not clear how this result was obtained and the comparison between the Raman signal of the bead residues and the one of modern madder roots, alizarine and purpurine is not entirely conclusive. The differences between the two types of signals should be discussed in more details. The reference signal of other red dyes could also be useful to compare with in order to rule out other hypotheses. Finally, I recommend a modification of the organisation of the paper. A section “material and methods” could be found before the presentation of the results. Otherwise, I have just minor comments that are detailed below. The paper is worth publishing with major revision.

Detailed comments

Title: 13 000 years ago instead of 15 000 years ago

Abstract and page1 line27-38 : red-pigment use started more than 300 000 ka in Africa. It predates modern human emergence in Kenya where it has been found. It is not unique to our species.

Brooks, A.S., Yellen, J.E., Potts, R., Behrensmeyer, A.K., Deino, A.L., Leslie, D.E., Ambrose, S.H., Ferguson, J.R., d’Errico, F., Zipkin, A.M., Whittaker, S., Post, J., Veatch, E.G., Foecke, K., Clark, J.B., 2018. Long-distance stone transport and pigment use in the earliest Middle Stone Age. Science 360, 90–94. https://doi.org/10.1126/science.aao2646

Abstract and page3 line91: “personal adornments” shouldn’t it be “personal ornaments”?

I would recommend to dedicate a special section for the presentation of the materials (line 120-129).

The method section should be placed before the presentation of the results.

In the method section the methods used to identify the molecules is not described. It is a key component of the paper, it must be presented.

Figure 5 – The main peak of the madder roots’ signal is not found on the bead residues’ signal. How do the authors explain such a difference?

Page 7 line243-249: This should be discussed in more details in the discussion section. Why do the authors think they identified madder dye in the residue on the beads, despite the differences observed between the ancient molecules and the modern madder roots? The reader needs to know more about the distinct hypotheses that could explain these differences.

Page7 line251: Discussion and conclusion

Page9 line304: The time gap between the previous earliest known use of madder dyes and the one described in this study is of 7,000 years only.

Page9 line329-330: carbon black could come from other organic sources than charcoal. These other sources should be cited too.

I would not say that the adding of carbon black increases the saturation of the ochre. I would rather think that it decreases it. For me, this argument is not valid. The adding of carbon black represents a manner of darkening the shade of the ochre.

Reviewer #2: I congratulate the authors to raise such an interesting topic.

However, i would recall that the use of natural dyes (and of pigments = of mineral origin) for their antibacterial and antifungal, preservation activities and in some cases even for medicinal purposes.

By adding in the Discussion considerations and an actual brief review of the above-mentioned alternative uses, it might add a more nuanced and pondered interpretation of the data that can drive toward a more complex involvement of the dyes in a far broader range of applications, escaping from the conventional framework of the "only" symbolic use.

Out of 16 analysed beads, 6 of them are covered with hematite (pigment). On the basis of preliminary SEM-EDS analysis, the remaining 10 red-stained shell beads were analyzed by Raman spectroscopy, more effective in identifying organic matter, and allowed to identify plant chromophores alizarin and purpurin. The data are interpreted as deriving from the processing of Rubia spp. to get the madder and identified plant source as Rubia tinctorum L..

In order increase the reproducibility of the research, it may be worth to consider other species while building the due reference collection - from which red dyes can be obtained –. For example, Alkanna tinctoria (alkanet or dyer's bugloss), Lawsonia inermis (henna) also present in the region and reported in Egyptian archaeology, Ferula assafoetida, various Galium various species (all available in the Mediterranean rim).

My apologies because the system do not maintain the latin name of the mentioned species.

Along the line of reproducibility, it would be worth that the authors report on their replicative experiment used to obtain the red madder under the conditions and the knowledge putatively convenient/available to the Natufians. To strengthen the interpretation it is also highly recommended to cross-check the replicative data with lab control of the analytical measurements with Standards (i.e. Sigma-Aldrich).

These are the main issues that I suggest to strength the impact of an otherwise very interesting research.

6. PLOS authors have the option to publish the peer review history of their article (what does this mean?). If published, this will include your full peer review and any attached files.

Reviewer #1: **Yes: **Laure Dayet

Reviewer #2: No

---

## [Author Response · Author response to Decision Letter 0]

15 Jul 2023

PONE-D-23-12821

Plant-based red colouration of shell beads 15,000 years ago in Kebara Cave, Mount Carmel (Israel)

Academic editor comments:

Thank you for submitting your manuscript to PLOS ONE. After careful consideration, we feel that it has merit but does not fully meet PLOS ONE’s publication criteria as it currently stands. Therefore, we invite you to submit a revised version of the manuscript that addresses the points raised during the review process.

This is an interesting study; however, both the reviewers expressed concerns regarding the identification of the materials purportedly used. For this contribution to be considered for publication, you will need to provide a full methodology, and all the relevant data, on an experimental programme covering a wide range of different materials and plants. You will then need to demonstrate, in detail, how you have reached your interpretations on the identification of the materials. The reader will need to be fully confident that they understand how you reached your results, and that, if required, this study is reproducible.

Reviewers' comments:

Reviewer #1: 

This manuscript presents the discovery of an organic plant-based dye on shell beads dating to about 13,000 years cal BP. This discovery is unique and predates the previously known evidence of plant-based dyes from 7,000 years. The data are well presented and the methodology is in adequation with the goals of the paper. However, if the presence of on organic dye is demonstrated, the interpretation that it comes from madder roots would benefit of further discussion. Indeed, it is not clear how this result was obtained and the comparison between the Raman signal of the bead residues and the one of modern madder roots, alizarine and purpurine is not entirely conclusive. The differences between the two types of signals should be discussed in more details.

Response:

The differences between archaeological and reference Raman signals is now detailed in p.12.

The reference signal of other red dyes could also be useful to compare with in order to rule out other hypotheses. 

Response:

Other red dyes of plant or animal origin are now detailed and shown in p.7-8, Tab.2 and compared to Kebara dye spectra in Fig.7.

Finally, I recommend a modification of the organisation of the paper. A section “material and methods” could be found before the presentation of the results. 

Response:

The description of SEM-EDS and Raman spectroscopy parameters detailed at the end of the manuscript in the “material and methods” section are very technical and we feel that, if they are moved before the results, it could interrupt article flow.

Otherwise, I have just minor comments that are detailed below. The paper is worth publishing with major revision.

Detailed comments

Title: 13 000 years ago instead of 15 000 years ago

Response: p.2 line 72-73 we wrote “layer B provided one of the oldest datings of the Natufian culture with a date of 12,470 ± 180 BP23 or 13,315-12,114 cal BC (calibrated in OxCal v.4.426).”

12,470 ± 180 BP is the uncalibrated date published by O. Bar-Yosef and A. Sillen (1993). The calibrated date being c. 13,000 before current era (13,315-12,114 cal BC) it means c.15,000 years ago (i.e., the beginning of the Natufian culture c. 15,000-11,650 cal BP).

Abstract and page1 line27-38 : red-pigment use started more than 300 000 ka in Africa. It predates modern human emergence in Kenya where it has been found. It is not unique to our species.

Brooks, A.S., Yellen, J.E., Potts, R., Behrensmeyer, A.K., Deino, A.L., Leslie, D.E., Ambrose, S.H., Ferguson, J.R., d’Errico, F., Zipkin, A.M., Whittaker, S., Post, J., Veatch, E.G., Foecke, K., Clark, J.B., 2018. Long-distance stone transport and pigment use in the earliest Middle Stone Age. Science 360, 90–94. https://doi.org/10.1126/science.aao2646

Response: Abstract p.1 line 18-19 we wrote “habitual use of red mineral pigments (such as iron-oxide e.g., ochre), by anatomically modern humans started in Africa about 140,000 years ago”

p.1 line 37-38 we wrote “anatomically modern humans probably explains, at least in part, why they started, about 140,000 years ago in Africa3, to use habitually red mineral pigments, such as iron-oxide”

We do not mean that the use of red pigments is unique to our species and that it started only 140,000 years ago. We mean (with reference to Wolf et al., 2018 p.186 “Habitual use of ochre starts about 140,000 years ago.”) that the habitual use (that Wolf et al., 2018 p.186 consider as “large assemblages ranging between hundreds and thousands of individual ochre pieces weighing a total of several kilograms”) of red pigments starts only 140,000 years ago and is associated to our species.

Wolf S., Conard N.J., Floss H., Dapschauskas R., Velliky E. and Kandel A.W. (2018). The Use of Ochre and Painting During the Upper Paleolithic of the Swabian Jura in the Context of the Development of Ochre Use in Africa and Europe. Open Archaeology, 4(1), pp.185-205. 

https://doi.org/10.1515/opar-2018-0012

Abstract and page3 line91: “personal adornments” shouldn’t it be “personal ornaments”?

Response: modified in the whole manuscript

I would recommend to dedicate a special section for the presentation of the materials (line 120-129).

Response: Table S1 in the supplementary materials which present the beads and pendants in detail (Type, sub-type, species, origin, type of identified pigment, position of the residues, etc) is now Table 1 in the main manuscript.

The method section should be placed before the presentation of the results.

In the method section the methods used to identify the molecules is not described. It is a key component of the paper, it must be presented.

Response:

The method used to identify the Natufian colourant are, as stated in the manuscript, Scanning Electron Microscopy with Energy Dispersive X-ray Spectroscopy (SEM-EDS) and Raman spectroscopy whose details of analysis are described in the method section. We added a description and a reference (n°85 and 86) in the method section to explain how those methods works:

85. Goldstein J., (2003). Scanning Electron Microscopy and X-Ray Microanalysis: Third Edition. Springer, 689p.

86. Ferraro J.R., (2003). Introductory Raman Spectroscopy. United Kingdom, Academic Press

Figure 5 – The main peak of the madder roots’ signal is not found on the bead residues’ signal. How do the authors explain such a difference?

Response:

We added several figures and sentences about this point (p.12-13):

“Also, despite being the closest match to the Natufian colourant (Fig.9), the Raman spectra from unprocessed roots of wild common madder (Rubia peregrina), slender-leaved madder (Rubia tenuifolia) and domesticated common madder (Rubia tinctorum) exhibit a strong signal at ca. 1290 cm-1 not shown in the Kebara spectra (Fig.9). This strong signal is not shown on the reference Madder Lake (a mixture of purpurin and alizarin made of R. tinctorum roots) (Fig.9) and therefore seems to correspond to transformation occurring during the many stages of transformation from fresh root to finished colourant, an observation confirmed by experimental manufacture of alizarin and purpurin lake46,72. A direct comparison of the Raman spectra of the Natufian colourant with reference molecules of Purpurin and Alizarin from Sigma Aldrich (Fig.10) show that many of the bands characteristic of this dye are also found in the spectra of these two main anthraquinones molecules originating from the Rubiaceae roots (Tab.2).”

Page 7 line243-249: This should be discussed in more details in the discussion section. Why do the authors think they identified madder dye in the residue on the beads, despite the differences observed between the ancient molecules and the modern madder roots? The reader needs to know more about the distinct hypotheses that could explain these differences.

Response: 

We added several sentences and figures (p.9-14):

“The closest match (without being a definite identification), was found with signatures from plant compounds, the anthraquinones (alizarin: C14H8O4 and purpurin: C14H8O5) isolated from the inner root parts of the Rubiaceae family (Rubia spp., Asperula spp., Gallium spp. (Tab.2))11.”

“With some representative Raman spectra obtained on the Kebara beads, Figure 8 shows the Raman spectra obtained on the main anthraquinones of Rubiaceae (purified dye of alizarin and purpurin from Sigma-Aldrich and Sennelier) isolated from the roots of domesticated common madder (Rubia tinctorum L.), a modern commercial madder lake (mixture of alizarin and purpurin from Kremer Pigmente) and directly measured on the coloured parts of the roots and berries (unprocessed) of wild common madder (Rubia peregrina L.) that we gathered near Paris, slender-leaved madder (Rubia tenuifolia L.) that we gathered near Jerusalem (Fig.S1) and domesticated common madder (Rubia tinctorum L.) bought from an herbalist. The Raman spectra of these products are different because of their nature, Rubia spp. roots being unprocessed and commercial references (Sennelier, Sigma-Aldrich and Kremer Pigmente (Fig.8)) resulting from a process of extraction/purification or synthesis that, obviously, would not have been possible in Prehistory. Similarities can be pointed out between Natufian colourant and modern references, even if there is no perfect match. Although the spectra obtained do not allow the origin of the Natufian colourant to be identified with certainty, everything suggests that it belongs to the family of anthraquinones derived from Rubiaceae plants. Bearing in mind that natural plant substances are complex mixtures of many molecules, differences could be explained by the characteristics of the Rubiaceae species that were accessible to the Natufians, other anthraquinone molecules such as pseudopurpurin, xanthopurpurin, rubiadin and munjistin could also be present in the Natufian colourant, the relative amounts of which vary with the age of the plant63. With regard to the madder plant, even when using a specific plant species, the final composition of a madder colourant can be largely affected by several factors such as cultivation parameters, harvesting and storage conditions of the roots64,65,66, effects of colorant extraction process (e.g., drying time, heating temperature, fermentation intensity)60 and the method used for colourant preparation64,67,68. In addition to the chemistry of the original products, time and the conservation environment may have modified the dye molecules. The taphonomic effects of the acidic conditions of Kebara cave sediment might have transformed the initial colourant (even if it preserved some colour and a molecular signature), knowing also that Raman spectra of dyes are strongly influenced by pH60,69,70. Furthermore, up to date, there is no other Raman reference for such an old complex organic dye that would allow us to make a comparison in a similar state of temporal degradation.

For example, the Raman spectra of reference purpurin from Sigma-Aldrich exhibit a strong signal at 1256 cm-1 (blue vertical dashed lines at the bottom of Figure 8), not shown on the spectra from Kebara beads, that is characteristic of pseudopurpurin71, an unstable anthraquinone that covert to purpurin (decarboxylation: loss of carbon dioxide) over time71 or depending on soaking and drying conditions of the roots60,72. Also, despite being the closest match to the Natufian colourant (Fig.9), the Raman spectra from unprocessed roots of wild common madder (Rubia peregrina), slender-leaved madder (Rubia tenuifolia) and domesticated common madder (Rubia tinctorum) exhibit a strong signal at ca. 1290 cm-1 not shown in the Kebara spectra (Fig.9). This strong signal is not shown on the reference Madder Lake (a mixture of purpurin and alizarin made of R. tinctorum roots) (Fig.9) and therefore seems to correspond to transformation occurring during the many stages of transformation from fresh root to finished colourant, an observation confirmed by experimental manufacture of alizarin and purpurin lake46,72. A direct comparison of the Raman spectra of the Natufian colourant with reference molecules of Purpurin and Alizarin from Sigma Aldrich (Fig.10) show that many of the bands characteristic of this dye are also found in the spectra of these two main anthraquinones molecules originating from the Rubiaceae roots (Tab.2). The mismatch in relative intensities, which is an essential feature of Raman signatures, does not allow formal identification, but these comparisons highlight the similarities that we have been able to underline.”

Page7 line251: Discussion and conclusion

Response: modified.

Page9 line304: The time gap between the previous earliest known use of madder dyes and the one described in this study is of 7,000 years only.

Response: As detailed in a previous response, the calibrated date being c. 13,000 before current era (13,315-12,114 cal BC) it means c. 15,000 years ago. Therefore, the time gap between the previous earliest known use of madder dyes (ca. 6,000 years ago9,10) and the one described in this study is of 9,000 years, that’s why we wrote “which are nearly 10,000 years older”. We modified the sentence for “older by 9,000 years from the first case of Rubiaceae use as a pigment published so far” to be more accurate. 

Page9 line329-330: carbon black could come from other organic sources than charcoal. These other sources should be cited too.

Response: modified for “Carbon black (produced, for example, by the incomplete combustion of vegetable matter)”

I would not say that the adding of carbon black increases the saturation of the ochre. I would rather think that it decreases it. For me, this argument is not valid. The adding of carbon black represents a manner of darkening the shade of the ochre.

Response: the sentence “the mixture of organic carbon black (produced, for example, by the combustion of vegetable matter) with the mineral ochre (which increases the saturation and darkness of the red)” has been changed for “the mixture of organic carbon black (produced, for example, by the combustion of vegetable matter) with the mineral ochre (which increases the darkness of the red)”.

Reviewer #2: 

I congratulate the authors to raise such an interesting topic.

However, i would recall that the use of natural dyes (and of pigments = of mineral origin) for their antibacterial and antifungal, preservation activities and in some cases even for medicinal purposes.

By adding in the Discussion considerations and an actual brief review of the above-mentioned alternative uses, it might add a more nuanced and pondered interpretation of the data that can drive toward a more complex involvement of the dyes in a far broader range of applications, escaping from the conventional framework of the "only" symbolic use.

Response:

We mention p.2 “Current Prehistoric research recognises red ochre as a universally applied material that serves a variety of different purposes from symbolic and ritual display to utilitarian or functional uses, depending on the contexts3,12.”

The context here being p.5 “the distribution of coloured residues in different parts of the beads suggests intentional colouration”

That is why we emphasize on the colouring purpose of the Natufian plant dye. We can discuss it in comparison with mineral pigment use for colouring purpose that is attested in the Natufian culture. It would be difficult to discuss medicinal purpose given that nothing is known in this field for the Natufian culture.

However, we mention in the discussion p.17 “the aerial parts of the wild madder, stems and leaves, have long been and are still used today in the traditional medicine of many cultures11,59,82,83 for their aphrodisiac, antibacterial, anti-oxidant, iron-rich properties, demonstrated through recent pharmaceutical studies84. If indeed we will be able to expand the research, in an endeavour to recover such findings through the study of other artefacts (e.g., groundstone vessels and tools), then we will enrich our knowledge of the Natufian culture and its practices in new, currently understudied, domains of existence.”

In summary, we do not rule out a use other than colouring, but further analysis is needed to go further in this field of research.

Out of 16 analysed beads, 6 of them are covered with hematite (pigment). On the basis of preliminary SEM-EDS analysis, the remaining 10 red-stained shell beads were analyzed by Raman spectroscopy, more effective in identifying organic matter, and allowed to identify plant chromophores alizarin and purpurin. The data are interpreted as deriving from the processing of Rubia spp. to get the madder and identified plant source as Rubia tinctorum L..

Response:

Plant source was identified as Rubia spp. We modified the manuscript to identify at the family level (Rubiaceae = Rubia spp., Asperula spp., Gallium spp.) given that we need invasive analysis (SERS or HPLC) in order to identify the actual plant species used by the Natufian (alizarin and purpurin anthraquinones are common to Rubia spp., Asperula spp., Gallium spp.). Invasive analysis was not authorized by Israel Antiquities Authority in this study, we hope that the publication of our article will make them reconsider these invasive analysis for a future study.

In order increase the reproducibility of the research, it may be worth to consider other species while building the due reference collection - from which red dyes can be obtained –. For example, Alkanna tinctoria (alkanet or dyer's bugloss), Lawsonia inermis (henna) also present in the region and reported in Egyptian archaeology, Ferula assafoetida, various Galium various species (all available in the Mediterranean rim).

My apologies because the system do not maintain the latin name of the mentioned species.

Response:

A list of potential plant and animal sources has been added in Table 2 and their chromophores Raman and SERS spectra compared to the characteristic bands of Natufian dye in Figure 7.

Along the line of reproducibility, it would be worth that the authors report on their replicative experiment used to obtain the red madder under the conditions and the knowledge putatively convenient/available to the Natufians. 

Response:

We did not reproduce red madder under the conditions and the knowledge putatively convenient/available to the Natufians given that, limited by non-invasive analytical methods, we have not been able to identify the actual plant species used by the Natufians. As stated in p.12 “With some representative Raman spectra obtained on the Kebara beads, Figure 8 shows the Raman spectra obtained on the main anthraquinones of Rubiaceae (purified dye of alizarin and purpurin from Sigma-Aldrich and Sennelier) isolated from the roots of domesticated common madder (Rubia tinctorum L.), a modern commercial madder lake (mixture of alizarin and purpurin from Kremer Pigmente) and directly measured on the coloured parts of the roots and berries (unprocessed) of wild common madder (Rubia peregrina L.) that we gathered near Paris, slender-leaved madder (Rubia tenuifolia L.) that we gathered near Jerusalem (Fig.S1) and domesticated common madder (Rubia tinctorum L.) bought from an herbalist. The Raman spectra of these products are different because of their nature, Rubia spp. roots being unprocessed and commercial references (Sennelier, Sigma-Aldrich and Kremer Pigmente (Fig.8)) resulting from a process of extraction/purification or synthesis that, obviously, would not have been possible in Prehistory.”

Also, p.16, we state that “Future studies of experimental manufacture of pigments from various Rubiaceae species coupled with invasive analytical methods on the archaeological beads and tools from Kebara could help identify the actual plant species used by the Natufians and its chaîne opératoire.”

To strengthen the interpretation it is also highly recommended to cross-check the replicative data with lab control of the analytical measurements with Standards (i.e. Sigma-Aldrich).

Response:

The spectra that we directly measured on the coloured parts of the roots and berries (unprocessed) of wild common madder (Rubia peregrina L.), slender-leaved madder (Rubia tenuifolia L.) and domesticated common madder (Rubia tinctorum L.) are cross-checked with standards from Sennelier, Sigma-Aldrich and Kremer Pigmente in Figure 8 and 9. The spectra of the Natufian dye is compared with these standards in Figure 10.

These are the main issues that I suggest to strength the impact of an otherwise very interesting research.

---

## [Editor Report · Decision Letter 1]

26 Jul 2023

PONE-D-23-12821R1Plant-based red colouration of shell beads 15,000 years ago in Kebara Cave, Mount Carmel (Israel)PLOS ONE

Dear Dr. Davin, 

Thank you for submitting your manuscript to PLOS ONE. After careful consideration, we feel that it has merit but does not fully meet PLOS ONE’s publication criteria as it currently stands. Therefore, we invite you to submit a revised version of the manuscript that addresses the points raised during the review process.

We look forward to receiving your revised manuscript.

Kind regards,

Karen Hardy

Academic Editor

PLOS ONE

Journal Requirements:

**Additional Editor Comments:**

The first sentence in the abstract is very broad. Human Universals have defined definitions.  I am not sure that this falls into that category. Please reword, to suggest that there is widespread evidence for decoration, or something similar.

I apologise that I did not notice reviewer’s Abstract and page3 line91: “personal adornments” shouldn’t it be “personal ornaments”? Response: modified in the whole manuscript.

You were correct and they are in fact, incorrect – it is adornment, not ornament.

Language errors have appeared where changes have been made. These all need to be corrected.

EG  41 . correct to plant or animal origin – this is ONE EXAMPLE only.  There are a lot more.

47 prehistoric (no capital)

55 ‘resting on a cushion’ - this needs further explanation. I do not understand what you mean here. What is the evidence for a ‘cushion’?

Your peaks don't align exactly. Raman is fine with one substance/source, but not good for complex organic mixtures See for example  Edwards, H.G. and Chalmers, J.M. eds., 2005. *Raman spectroscopy in archaeology and art history* (Vol. 9). Royal Society of Chemistry.

If this is just madder (Rubia spp.) there is arguably a reasonable chance of identifying it using Raman, but even alizarin and purpurin can be difficult to distinguish from each other, unless the pH is fairly alkaline. You have bands which fit with purpurin and alizarin - a mixture of the two - but several of the peaks don't align exactly. There may be reasons for this and the Israeli madder may well be Rubia tinctorum but this is based largely on context and inference rather than a high level of correlation. 

To be suitable for publication you need to add more important qualifying statements.  You cannot go further than suggesting that this might be…. Etc.

For example: Line 157 –Change to ‘suggest’ the presence of…

---

## [Author Response · Author response to Decision Letter 1]

2 Sep 2023

Journal Requirements:

Response:

The reference list is complete and correct. All the references are cited in the text, the figure captions and the method section.

Academic editor comments:

The first sentence in the abstract is very broad. Human Universals have defined definitions. I am not sure that this falls into that category. Please reword, to suggest that there is widespread evidence for decoration, or something similar.

Response:

“Decorating the living space, objects, body and clothes with colour seems to be a universal human practice” is now “Decorating the living space, objects, body and clothes with colour is a widespread human practice”

I apologise that I did not notice reviewer’s Abstract and page3 line91: “personal adornments” shouldn’t it be “personal ornaments”? Response: modified in the whole manuscript.

You were correct and they are in fact, incorrect – it is adornment, not ornament.

Response:

Modified in the whole manuscript

Language errors have appeared where changes have been made. These all need to be corrected.

EG 41 . correct to plant or animal origin – this is ONE EXAMPLE only. There are a lot more.

47 prehistoric (no capital)

Response:

Modified in the whole manuscript

55 ‘resting on a cushion’ - this needs further explanation. I do not understand what you mean here. What is the evidence for a ‘cushion’?

Response:

According to Fanny Bocquentin's observations during the excavation of Raquefet, the movement of the skeleton in the sepulchral space after the decomposition of the body indicates that in some cases the head of the deceased was placed on a cushion made of perishable material that was coloured red. We have removed this sentence from the manuscript as these observations have not yet been published. F. Bocquentin will be writing a detailed article on the subject in the future.

Your peaks don't align exactly. Raman is fine with one substance/source, but not good for complex organic mixtures See for example Edwards, H.G. and Chalmers, J.M. eds., 2005. Raman spectroscopy in archaeology and art history (Vol. 9). Royal Society of Chemistry.

If this is just madder (Rubia spp.) there is arguably a reasonable chance of identifying it using Raman, but even alizarin and purpurin can be difficult to distinguish from each other, unless the pH is fairly alkaline. You have bands which fit with purpurin and alizarin - a mixture of the two - but several of the peaks don't align exactly. There may be reasons for this and the Israeli madder may well be Rubia tinctorum but this is based largely on context and inference rather than a high level of correlation. 

To be suitable for publication you need to add more important qualifying statements. You cannot go further than suggesting that this might be…. Etc.

Response:

We do not state that we identify the exact plant species that the Natufian used. We state that the Natufian organic colourant is made of a mixture that contain alizarin and purpurin from the roots of one of the Rubiaceae family plants (Rubia spp., Asperula spp., Gallium spp.): 

Line 221 “The closest match (without being a definite identification), was found with signatures from plant compounds, the anthraquinones (alizarin: C14H8O4 and purpurin: C14H8O5) isolated from the inner root parts of the Rubiaceae family (Rubia spp., Asperula spp., Gallium spp. (Tab.2))”

Line 264-267 “Similarities can be pointed out between Natufian colourant and modern references, even if there is no perfect match. Although the spectra obtained do not allow the origin of the Natufian colourant to be identified with certainty, everything suggests that it belongs to the family of anthraquinones derived from Rubiaceae plants.”

Line 296-298 “The mismatch in relative intensities, which is an essential feature of Raman signatures, does not allow formal identification, but these comparisons highlight the similarities that we have been able to underline between the Natufian organic colourant and compounds from Rubiaceae plants.” We added this final part to the sentence.

Line 367-369 “Future studies of experimental manufacture of pigments from various Rubiaceae species coupled with invasive analytical methods on the archaeological beads and tools from Kebara could help identify the actual plant species used by the Natufians and its chaîne opératoire.”

For example: Line 157 –Change to ‘suggest’ the presence of…

Response:

Line 157 “Raman spectra (λexc = 458 nm) obtained on Kebara Cave Early Natufian beads and pendants highlighting the presence of hematite” is about Fig.3 where archaeological Raman peaks align exactly with reference hematite peaks. In that case, the identification is clear.

---

## [Editor Report · Decision Letter 2]

11 Sep 2023

PONE-D-23-12821R2Plant-based red colouration of shell beads 15,000 years ago in Kebara Cave, Mount Carmel (Israel)PLOS ONE

Dear Dr. Davin,

Thank you for submitting your manuscript to PLOS ONE. After careful consideration, we feel that it has merit but does not fully meet PLOS ONE’s publication criteria as it currently stands. Therefore, we invite you to submit a revised version of the manuscript that addresses the points raised during the review process.

There continue to be language issues, please check your ms again and correct these.  I asked that you qualify some of your statements.  While I agree that in many cases you are in general not being definitive, you need to change in every case, use of the word 'highlight(ing)' to 'suggest(ing)'.  There is a very clear difference; highlighting leaves no room for the possibility that it could be incorrect while 'suggesting' does. 

We look forward to receiving your revised manuscript.

Kind regards,

Karen Hardy

Academic Editor

PLOS ONE
---

## [Author Response · Author response to Decision Letter 2]

14 Sep 2023

Academic editor comments:

There continue to be language issues, please check your ms again and correct these. 

Response:

Language issues (correctness and clarity) have been fixed in the manuscript.

I asked that you qualify some of your statements. While I agree that in many cases you are in general not being definitive, you need to change in every case, use of the word 'highlight(ing)' to 'suggest(ing)'. There is a very clear difference; highlighting leaves no room for the possibility that it could be incorrect while 'suggesting' does.

Response:

'highlight(ing)' has been changed to 'suggest(ing)' in the whole manuscript.

---

## [Editor Report · Decision Letter 3]

18 Sep 2023

Plant-based red colouration of shell beads 15,000 years ago in Kebara Cave, Mount Carmel (Israel)

PONE-D-23-12821R3

Dear Dr. Davin,

We’re pleased to inform you that your manuscript has been judged scientifically suitable for publication and will be formally accepted for publication once it meets all outstanding technical requirements.

Kind regards,

Karen Hardy

Academic Editor

PLOS ONE
---

## [Editor Report · Acceptance letter]

4 Oct 2023

PONE-D-23-12821R3 

Plant-based red colouration of shell beads 15,000 years ago in Kebara Cave, Mount Carmel (Israel) 

Dear Dr. Davin:

I'm pleased to inform you that your manuscript has been deemed suitable for publication in PLOS ONE. Congratulations! Your manuscript is now with our production department. 

Kind regards, 

on behalf of

Dr. Karen Hardy 

Academic Editor

PLOS ONE